# Strong-field quantum control in the extreme ultraviolet domain using pulse shaping

Fabian Richter[1], Ulf Saalmann[2], Enrico Allaria[3], Matthias Wollenhaupt[4], Benedetto Ardini[5], Alexander Brynes[3], Carlo Callegari[3], Giulio Cerullo[5], Miltcho Danailov[3], Alexander Demidovich[3], Katrin Dulitz[6], Raimund Feifel[7], Michele Di Fraia[3,8], Sarang Dev Ganeshamandiram[1], Luca Giannessi[3,9], Nicolai Gölz[1], Sebastian Hartweg[1], Bernd von Issendorff[1], Tim Laarmann[10,11], Friedemann Landmesser[1], Yilin Li[1], Michele Manfredda[3], Cristian Manzoni[12], Moritz Michelbach[1], Arne Morlok[1], Marcel Mudrich[13], Aaron Ngai[1], Ivaylo Nikolov[3], Nitish Pal[3], Fabian Pannek[14], Giuseppe Penco[3], Oksana Plekan[3], Kevin C. Prince[3], Giuseppe Sansone[1], Alberto Simoncig[3], Frank Stienkemeier[1], Richard James Squibb[7], Peter Susnjar[3], Mauro Trovo[3], Daniel Uhl[1], Brendan Wouterlood[1], Marco Zangrando[3,8] & Lukas Bruder[1]✉

Tailored light–matter interactions in the strong coupling regime enable the manipulation and control of quantum systems with up to unit efficiency[1,2], with applications ranging from quantum information to photochemistry[3–7]. Although strong light–matter interactions are readily induced at the valence electron level using long-wavelength radiation[8], comparable phenomena have been only recently observed with short wavelengths, accessing highly excited multi-electron and inner-shell electron states[9,10]. However, the quantum control of strong-field processes at short wavelengths has not been possible, so far, because of the lack of pulse-shaping technologies in the extreme ultraviolet (XUV) and X-ray domain. Here, exploiting pulse shaping of the seeded free-electron laser (FEL) FERMI, we demonstrate the strong-field quantum control of ultrafast Rabi dynamics in helium atoms with high fidelity. Our approach reveals a strong dressing of the ionization continuum, otherwise elusive to experimental observables. The latter is exploited to achieve control of the total ionization rate, with prospective applications in many XUV and soft X-ray experiments. Leveraging recent advances in intense few-femtosecond to attosecond XUV to soft X-ray light sources, our results open an avenue to the efficient manipulation and selective control of core electron processes and electron correlation phenomena in real time.

Strong-field phenomena play an important part in our understanding of the quantum world. Light–matter interactions beyond the perturbative limit can substantially distort the energy landscape of a quantum system, which forms the basis of many strong-field effects[8] and provides opportunities for efficient quantum control schemes[11]. Moreover, resonant strong coupling induces rapid Rabi cycling of the level populations[12], enabling complete population transfer to a target state[2]. The development of intense extreme ultraviolet (XUV) and X-ray light sources has recently led to the investigation of related phenomena beyond valence electron dynamics, in highly excited, multi-electron and inner-shell electron states[9,10,13–17]. Yet in most of these studies, the dressing of the quantum systems was induced by intense infrared fields overlapping with the XUV and X-ray pulses. In contrast, the alteration of energy levels directly by short-wavelength radiation is more difficult. So far, only a few studies have reported XUV-induced AC-Stark shifts of moderate magnitude (≲100 meV), difficult to resolve experimentally[9,18–20].

Another important step in exploring and mastering the quantum world is the active control of quantum dynamics with tailored light fields[21–23]. At long wavelengths, sophisticated pulse-shaping techniques facilitate the precise quantum control and even the adaptive-feedback control of many light-induced processes, in both weak- and strong-field regimes[24–28]. Several theoretical studies have pointed out the potential of pulse shaping in XUV and X-ray experiments[29–31]. As an experimental step in this direction, phase-locked monochromatic and polychromatic pulse sequences have been generated[32–35]. Using this tool, coherent control demonstrations in the perturbative limit[32,35,36] and the generation of intense attosecond pulses were achieved[37]. Moreover, ultrafast polarization shaping at XUV wavelengths[38] and chirp control for the temporal compression of XUV pulses[39] were recently demonstrated. However, spectral phase shaping, which forms the core of pulse-shaping techniques, has not been demonstrated for the control of quantum phenomena at short wavelengths. Here we establish spectral phase

[1]Institute of Physics, University of Freiburg, Freiburg, Germany. [2]Max-Planck-Institut für Physik komplexer Systeme, Dresden, Germany. [3]Elettra-Sincrotrone Trieste S.C.p.A., Trieste, Italy. [4]Institute of Physics, University of Oldenburg, Oldenburg, Germany. [5]Dipartimento di Fisica, IFN-CNR, Milan, Italy. [6]Institut für Ionenphysik und Angewandte Physik, Universität Innsbruck, Innsbruck, Austria. [7]Department of Physics, University of Gothenburg, Gothenburg, Sweden. [8]Istituto Officina dei Materiali, CNR (CNR-IOM), Trieste, Italy. [9]Istituto Nazionale di Fisica Nucleare, Laboratori Nazionali di Frascati, Frascati, Italy. [10]Deutsches Elektronen-Synchrotron DESY, Hamburg, Germany. [11]The Hamburg Centre for Ultrafast Imaging CUI, Hamburg, Germany. [12]IFN-CNR, Milan, Italy. [13]Department of Physics and Astronomy, Aarhus University, Aarhus, Denmark. [14]Institute for Experimental Physics, University of Hamburg, Hamburg, Germany. ✉e-mail: lukas.bruder@physik.uni-freiburg.de

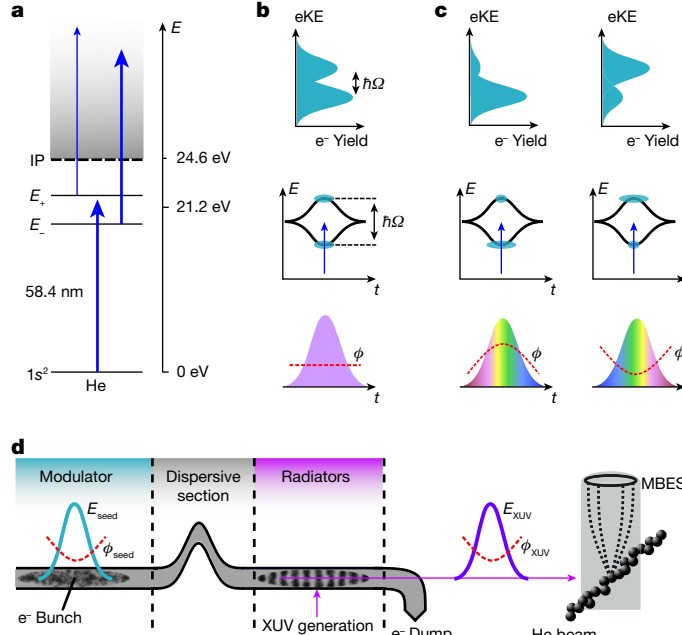

**Fig. 1 | XUV strong-field coherent control scheme. a**, Intense XUV pulses dress the He $1s^2$, $1s2p$ states and the electron continuum. $E_\pm$ labels indicate the bound dressed states correlated to the $1s2p$ bare state. Mixing of p- and d-waves in the dressed continuum results in different coupling strengths to the dressed bound states (indicated by the thickness of the arrows). **b,c**, In the time domain, the AT splitting follows the intensity profile of the XUV field (middle). The dressed-state populations are monitored in the photoelectron eKE distributions (top). XUV pulse shaping enables the control of the non-perturbative quantum dynamics (bottom). For a flat phase $\phi$ (no chirp), both the excited dressed states are equally populated. For a positive phase curvature (up chirp), the population is predominantly transferred to the lower dressed state and the upper state is depleted, whereas for negative curvature (down chirp), the situation is reversed. **d**, Principle of XUV pulse shaping at the FEL FERMI. Intense seed laser pulses overlap spatially and temporally with the relativistic electron bunch in the modulator section of the FEL, leading to a modulation in the electron phase space. The induced energy modulations are converted into electron-density oscillations on passing a dispersive magnet section. The micro-bunched electrons then propagate through a section of radiator undulators, producing a coherent XUV pulse. In this process, the phase function of the seed pulse is coherently transferred to the XUV pulse, resulting in precise XUV phase shaping. The FEL pulses are focused on the interaction volume, exciting and ionizing He atoms. The photoelectrons are detected with a magnetic bottle electron spectrometer (MBES).

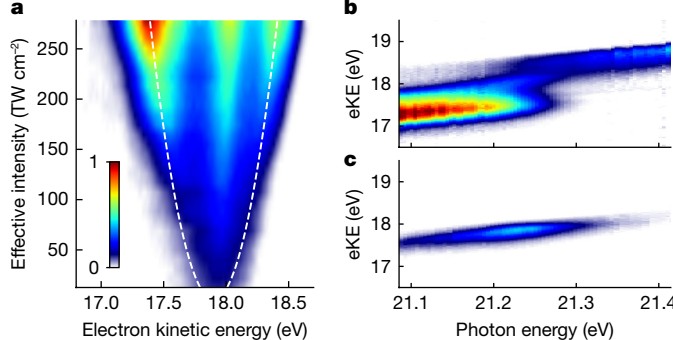

**Fig. 2 | Build-up of the AT splitting in He atoms. a**, Detected photoelectron eKE distribution (raw data) as a function of the XUV intensity (FEL photon energy: 21.26 eV, GDD = 135 fs²). Dashed lines show the calculated AT splitting for an effective XUV peak intensity $I_{eff}$ accounting for the spatial averaging in the interaction volume. **b,c**, Photoelectron spectra as a function of photon energy recorded for high XUV intensity ($I_{eff} = 2.92(18) \times 10^{14}$ W cm⁻²) (**b**) and for lower intensity ($I_{eff} \approx 10^{13}$ W cm⁻²) (**c**). In **b**, an avoided crossing between the lower and higher AT band is visible directly in the raw photoelectron spectra. The photoelectron distribution peaking at eKE = 17.9 eV in **a** and **b** is ascribed to He atoms excited by lower XUV intensity (see text).

shaping of intense XUV laser pulses and demonstrate high-fidelity quantum control of the Rabi and photoionization dynamics in helium.

In the experiment, He atoms are dressed and ionized by intense coherent XUV pulses ($I > 10^{14}$ W cm⁻²) delivered by the seeded FEL FERMI (Fig. 1a). The high radiation intensity causes a strong dressing of both the bound states in He and the photoelectron continuum, whereas the dynamics of the quantum system are still in the multiphoton regime (Keldysh parameter $\gamma = 11$). By contrast, the dynamics of a system dressed with near-infrared (NIR) radiation of comparable intensity would be dominated by tunnel and above barrier ionization ($\gamma = 0.35$) (ref. 8). Hence, the use of short-wavelength radiation provides access to a unique regime, in which the interplay between strongly dressed bound states and a strongly dressed continuum can be studied.

To dress the He atoms, we induce rapid Rabi cycling of the $1s^2 \to 1s2p$ atomic resonance with a near-resonant field $E(t)$. The generalized Rabi frequency of this process is $\Omega = \hbar^{-1}\sqrt{(\mu E)^2 + \delta^2}$, where $\mu$ denotes the transition dipole moment of the atomic resonance, $\delta$ the energy detuning and $\hbar$ the reduced Planck constant. In the dressed-state formalism,

the eigenenergies of the bound states depend on the field intensity and show the characteristic Autler–Townes (AT) energy splitting $\Delta E = \hbar\Omega$ (ref. 40). The observation of this phenomenon requires the mapping of the transiently dressed level structure of He while perturbed by the external field[41]. This is achieved by immediate photoionization over the course of the femtosecond pulses, thus projecting the time-integrated energy level shifts onto the electron kinetic energy (eKE) distribution (Fig. 1b).

Analogous to the bound-state description, the dressed continuum states are obtained by diagonalization of the corresponding Hamiltonian. The hybrid electron–photon eigenstates consist of a mixing of partial waves with different angular momenta, which alters the coupling strength to the dressed bound states of the He atoms (Fig. 1a).

Figure 2 demonstrates experimentally the dressing of the He atoms. The build-up of the AT doublet is visible in the raw photoelectron spectra as the XUV intensity increases (Fig. 2a). The evolution of the AT doublet splitting is in good agreement with the expected square-root dependence on the XUV intensity $\Delta E = \mu\sqrt{2I_{eff}/(\epsilon_0 c)}$. Here, $I_{eff}$ denotes an effective peak intensity, accounting for the spatially averaged intensity distribution in the interaction volume, $\epsilon_0$ denotes the vacuum permittivity and $c$ denotes the speed of light. The data can be thus used for gauging the XUV intensity in the interaction volume, a parameter otherwise difficult to determine. At the maximum XUV intensity, the photoelectron spectrum shows an energy splitting exceeding 1 eV, indicative of substantial AC-Stark shifts in the atomic level structure. The large AT splitting further implies that a Rabi flopping within 2 fs is achieved, offering a perspective for rapid population transfer outpacing possible competing intra- and inter-atomic decay mechanisms, which are ubiquitous in XUV and X-ray applications.

Figure 2b,c shows the photoelectron yield as a function of excitation photon energy. For high XUV intensity (Fig. 2b), the photoelectron spectra show an avoided level crossing of the dressed He states as they are mapped to the electron continuum (see also Fig. 4). Accordingly, at lower XUV intensity (Fig. 2c), the avoided crossing is not visible anymore. In the latter, the eKE distribution centres at 17.9 eV. In Fig. 2b, a similar contribution appears at the same kinetic energy that overlays the photoelectrons emitted from the strongly dressed atoms. Likewise, a notable portion of photoelectrons at eKE ≈ 17.9 eV in Fig. 2a does not show a discernible AT splitting. We conclude that a fraction of He atoms in the ionization volume are excited by much lower FEL intensity, which is consistent with the aberrated intensity profile of the FEL measured

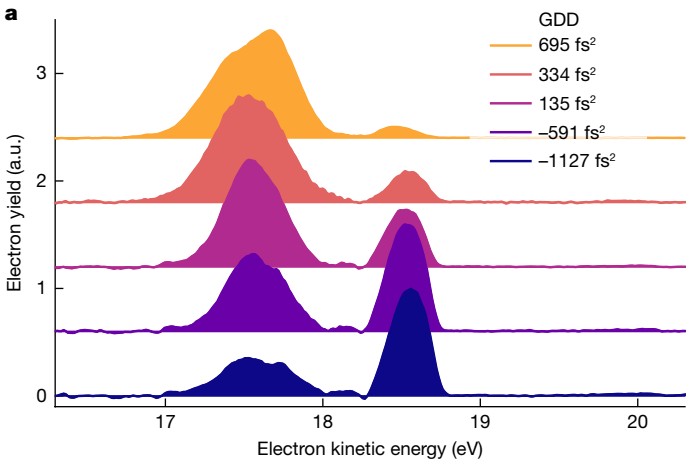

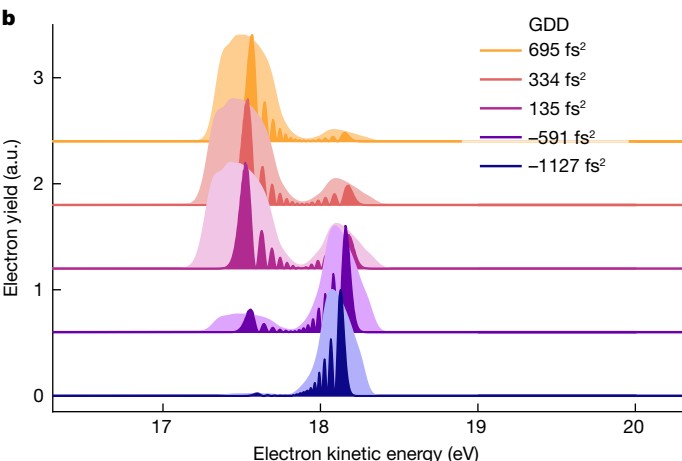

**Fig. 3 | Strong-field quantum control of dressed He populations.**
**a**, Photoelectron spectra obtained for phase-shaped XUV pulses (see labels for GDD values; photon energy = 21.25 eV; $I_{eff}$ = 2.8(2) × 1,014 W cm⁻²). The control of the dressed-state populations is directly reflected in the relative change of amplitude in the photoelectron bands. The small peak at 18.13 eV results from imperfect removal of the lower intensity contribution from the aberrated focus. **b**, Calculations of the time-dependent Schrödinger equation for a single

active electron (TDSE-SAE) and a single laser intensity corresponding to the experimental $I_{eff}$ = 2.8 × 10¹⁴ W cm⁻² (dark colours). Spectral fringes reflect here the temporal progression of the Rabi frequency during the light–matter interaction. The broadened photoelectron spectra (light colours) account for experimental broadening effects caused by the focal intensity averaging and the instrument response function. a.u., arbitrary units.

in the ionization volume (Extended Data Fig. 1). This overlapping lower intensity contribution does not influence the interpretation of the results in this work. For better visibility of the main features, we thus subtract this contribution from the data shown in Figs. 3 and 4.

The demonstrated dressing of He atoms provides the prerequisite for implementing the strong-field quantum control scheme (Fig. 1b,c). The main mechanism underlying the control scheme is described in the framework of the selective population of dressed states (SPODS), which is well established in the NIR spectral domain[28]. Here, we extend SPODS to the XUV domain and include a new physical aspect—that is, the transition of the bound atomic system into a strongly dressed continuum. In SPODS, a flat phase leads to an equal population of both dressed states in the excited state manifold of helium; a positive phase curvature results in a predominant population of the lower dressed state and a negative phase curvature results in a predominant population of the upper dressed state (Fig. 1c). The scheme has been experimentally demonstrated with long-wavelength radiation[42], in which pulse-shaping techniques are readily available. However, the opportunities for pulse-shaping technologies are largely unexplored for XUV and X-ray radiation.

We solve this problem by exploiting the potential of seeded FELs to allow for the accurate control of XUV pulse properties[39,43]. These demonstrations have been so far limited to applications of temporal compression and amplification of the FEL pulses. By contrast, the deterministic control of quantum dynamics in a material system involves many more degrees of freedom, which makes the situation considerably more complex. The seeded FEL FERMI operation is based on the high-gain harmonic generation (HGHG) principle[44], in which the phase of an intense seed laser pulse is imprinted into a relativistic electron pulse to precondition the coherent XUV emission at harmonics of the seed laser (Fig. 1d). For FEL operation in the linear amplification regime, the phase $\phi_{nH}(t)$ of the FEL pulses emitted at the $n$'th harmonic of the seed laser follows the relationship[39]

$$\phi_{nH}(t) \approx n[\phi_s(t) + \phi_e(t)] + \phi_a. \tag{1}$$

Here, $\phi_s$ denotes the phase of the seed laser pulses, which can be tuned with standard pulse-shaping technology at long wavelengths (Methods); $\phi_e$ accounts for the possible phase shifts caused by the energy dispersion of the electron beam through the dispersive magnet and is

negligible for the parameters used in the experiment; and $\phi_a$ accounts for the FEL phase distortion due to the amplification and saturation in the radiator and has been kept negligibly small by properly tuning the FEL (Methods). Although complex phase shapes may be implemented with this scheme, for the current objective of controlling the strong-field induced dynamics in He atoms, shaping the quadratic phase term (group delay dispersion (GDD)) is sufficient[42]. Therefore, we focus on the GDD control in the following discussion.

Figure 3 demonstrates the quantum control of the dressed He populations. The eKE distribution shows a pronounced dependence on the GDD of the XUV pulses (Fig. 3a). At minimum chirp (GDD = 135 fs²), we observe an almost even amplitude in the AT doublet, whereas for GDD < 0, the higher energy photoelectron band dominates; for GDD > 0, the situation is reversed. These changes directly reflect the control of the relative populations in the upper and lower dressed states of the He atoms. We obtain an excellent control contrast and the results are highly robust (Extended Data Fig. 2), which is remarkable given the complex experimental setup.

The experiment is in good agreement with the theoretical model (Fig. 3b) numerically solving the time-dependent Schrödinger equation for a single active electron (TDSE-SAE; Methods). To account for experimental broadening effects, we calculated the photoelectron spectra for a single intensity (corresponding to the experimental $I_{eff}$) and including the focal intensity average present in the experiment (Methods). All salient features of the experiment are well reproduced. The control of the dressed-state populations is in very good qualitative agreement. The different widths and shapes of the photoelectron peaks are qualitatively well-matched between the experiment and the calculations. The difference in the AT energy splitting between the experiment ($\Delta E_{exp} \approx 1.02$ eV) and theory ($\Delta E_{theo} = 0.74$ eV) is in good agreement with the fact that the model underestimates the transition dipole moment of the $1s^2 \rightarrow 1s2p$ transition by a factor of 1.4 (Methods).

The high reproducibility, the excellent control contrast and the good agreement with theory confirm the feasibility of precise pulse shaping in the XUV domain and of quantum control applications, even of transient strong-field phenomena. This is an important achievement in view of quantum optimal control applications at short wavelengths.

The implemented control scheme is not restricted to adiabatic processes[28]. In our experiment, the dynamics are adiabatic only for the largest frequency chirp (GDD = −1,127 fs²) (Extended Data Fig. 3). However,

this also shows that the condition for rapid adiabatic passage[2] can be generally reached with our approach, offering a perspective on efficient population transfer in the XUV and potentially in the soft X-ray regime.

The active control of quantum dynamics with tailored light fields is an asset of pulse shaping. As another asset, systematic studies with shaped laser pulses can be used to uncover underlying physical mechanisms that are otherwise hidden. Here, we demonstrate this concept for pulse shaping in the XUV domain. The high XUV intensities used in our study lead to a peculiar scenario in which both bound and continuum states are dressed and a complex interplay between their dynamics arises. Hence, for a comprehensive understanding of the strong-field physics taking place, the bound-state dynamics and the non-perturbative photoionization have to be considered. This is in contrast to the strong-field control at long wavelengths, for which the continuum could be described perturbatively[42].

Figure 4a,b shows the avoided crossing of the photoelectron bands for different spectral phase curvatures applied to the XUV pulses. The experimental data show a clear dependence of the AT doublet amplitudes on the detuning and the GDD of the driving field, in good agreement with the theory. In the strong dressing regime, the bound–continuum coupling marks a third factor that influences the photoelectron spectrum. As predicted by theory, the strong-field-induced mixing of continuum states (Fig. 1a) leads to different photoionization probabilities for the upper and lower dressed states of the bound system[45]. This is in agreement with the prevalent asymmetry of the AT doublet amplitudes observed in our data and calculations (Fig. 4a,b). An analogous effect is observed for the strong-field bound–continuum coupling in solid state systems[46].

To disentangle this strong-field effect from the influence of the detuning and spectral phase of the driving field, we evaluate the amplitude ratio between the upper and lower photoelectron bands at detuning $\delta = 0$ eV (Fig. 4c). Interpolation to GDD = 0 fs$^2$ isolates the asymmetry solely caused by the strong-field bound–continuum coupling. We find reasonable agreement with our model when including the dressing of the ionization continuum (blue curve), in stark contrast to the same model but treating the continuum perturbatively (yellow curve). Hence, the dressing of the He atoms provides a probe of the strong-field dynamics in the continuum. This property is otherwise difficult to access and becomes available through our systematic study of the spectral phase dependence on the photoelectron spectrum.

Another possible mechanism for a general asymmetry in the AT doublet amplitudes could be the interference between ionization pathways through resonant and near-resonant bound states as recently suggested for the dressing of He atoms with XUV[20,47] and for alkali atoms with bichromatic NIR fields[48]. In our experiment, we study the energetically well-isolated transition $1s^2 \rightarrow 1s2p$, in which the contributions from neighbouring optically active states should be negligible. This provides us with a clean two-level system and greatly simplifies the data interpretation. For confirmation, we performed a calculation with a modified model in which any two-photon ionization through near-resonant states (except for the $1s2p$ state) was suppressed and, thus, possible photoionization interference effects were eliminated. Still, we observe a pronounced asymmetry in the AT doublet amplitudes (Extended Data Fig. 4). Moreover, owing to the large Keldysh parameter ($\gamma = 11$) and the low ponderomotive potential ($U_p < 100$ meV) in our study, other strong-field effects are expected to play a negligible part in the observed dynamics. We thus assign the experimental observation to the coupling of the dressed atom dynamics with a dressed ionization continuum induced by intense XUV driving fields.

A comprehensive understanding of the strong-field-induced dynamics in the system lays the basis for another quantum control effect, that is, the suppression of the ionization rate of the system, as proposed theoretically[45]. The excitation probability for one-photon transitions is generally independent of the chirp direction of the driving field. However, if driving a quantum system in the strong-field limit, its

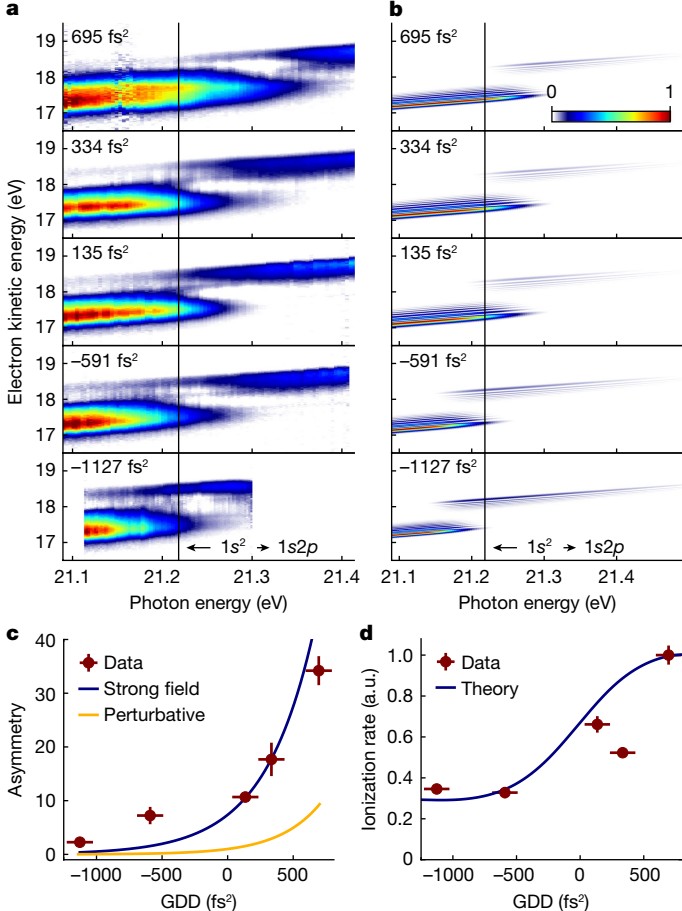

**Fig. 4 | Energy-domain representation of the quantum control scheme.** **a**, Photoelectron spectra as a function of energy detuning for different GDD values as labelled ($I_{eff} = 2.92(18) \times 10^{14}$ W cm$^{-2}$). **b**, TDSE-SAE calculations. Broadening by the instrument response function is omitted in the model. **c**, Amplitude ratio between the upper and lower photoelectron bands evaluated at the $1s^2 \rightarrow 1s2p$ resonance; hence, $\delta = 0$. Experimental data (red), TDSE-SAE model treating the bound and continuum dynamics non-perturbatively (blue) and TDSE-SAE model applied to the bound-state dynamics, but treating the continuum perturbatively (yellow). **d**, Dependence of the He ionization rate on the spectral phase of the driving field. Data (red) and TDSE-SAE model (blue). a.u., arbitrary units.

quasi-resonant two-photon ionization rate may become sensitive to the chirp direction. We demonstrate the effect experimentally in Fig. 4d. A substantial reduction of the He ionization rate by 64% is achieved, solely by tuning the chirp of the FEL pulses while keeping the pulse area constant. The good agreement with the TDSE-SAE calculations confirms the mechanism. This control scheme exploits the interplay between the bound-state dynamics and the above-discussed selective coupling of the upper and lower dressed states to the ionization continuum. We note a stabilization mechanism of the dressed states in He was recently proposed, effectively causing also a suppression of the ionization rate[47]. This mechanism requires, however, extreme pulse parameters, difficult to achieve experimentally. By contrast, our approach based on shaped pulses is more feasible and applies to a broader parameter range.

With this work, we have established a new tool for the manipulation and control of matter using XUV light sources. The demonstrated concept offers a wide pulse shaping window regarding pulse duration, photon energy and more complex phase shapes. In particular, the recent progress in echo-enabled harmonic generation[49,50] promises to extend the pulse-shaping concept to the soft X-ray domain (up to the 600 eV range) in which localized core electron states can be addressed.

As such, we expect our work will stimulate other experimental and theoretical activities exploring the exciting possibilities offered by XUV and soft X-ray pulse shaping: first theory proposals in this direction have already been made[29–31]. The demonstrated scheme already sets the basis for highly efficient adiabatic population transfer[1,2] and an extension to cubic or sinusoidal phase shaping would open up many more interesting control schemes[26,27]. This may find applications, for example, in valence-core-stimulated Raman scattering or efficient and fast qubit manipulation with XUV and soft X-ray light. Furthermore, selective control schemes may reduce the influence of competing ionization processes ubiquitous in XUV and X-ray spectroscopy and imaging experiments, for which our work provides experimental demonstration. The generation of coherent attosecond pulse trains, with independent control of amplitude and phases, has been demonstrated at seeded FELs[37], bringing pulse shaping applications on the attosecond time scale within reach. This paves the way for the quantum control of molecular and solid state systems with chemical selectivity and on attosecond time scales.

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

## Methods

### Experiment

The experiments were performed at the low-density matter endstation[51] of the FEL FERMI-1 (ref. 44). The FEL was operated in circular polarization at the sixth harmonic of the seed laser. The FEL photon energy was tuned in the 21.05–21.47 eV range with an optical parametric amplifier in the seed laser setup. The maximum pulse energy at the target was $E_{max}$ = 71 µJ, taking transmission losses into account. A $N_2$-gas filter was used for continuous attenuation of the pulse energy. For the data in Fig. 2c, an Sn filter (thickness 200 nm) was inserted, attenuating the XUV intensity by roughly one order of magnitude. At minimum chirp setting (GDD = 135 fs$^2$), an FEL pulse duration of 49(3) fs was measured by a cross-correlation between the FEL pulses and an 800-nm auxiliary pulse. The beam size at the target was 8.00(8) × 11.3(1) µm$^2$, reconstructed with a Hartmann wavefront sensor. Assuming a Gaussian spatial mode, this yields a calculated estimate for the maximum reachable peak intensity of 3.84 × 10$^{14}$ W cm$^{-2}$ at the interaction region. In comparison, the effective intensity deduced from the AT splitting is $I_{eff}$ = 2.78(2) × 10$^{14}$ W cm$^{-2}$. This value is 27% smaller than the value calculated for a Gaussian spatial mode, hinting at an aberrated spatial mode (see also Extended Data Fig. 1).

Spectral phase shaping of the seed laser was implemented by tuning a single-pass transmission grating compressor and characterized by self-diffraction frequency-resolved optical gating. In the applied tuning range, changes in higher-order phase terms are small and are thus neglected. The coherent transfer of the seed phase $\phi_s$ to the FEL phase $\phi_{nH}$ was calculated with FEL simulations using the GENESIS 1.3 code[52] and an FEL model[53]. With these tools, the FEL was analysed as outlined in ref. 39 for a set of seed laser and FEL settings before the beamtime. Details can be found in ref. 53. To minimize the additional chirp introduced by the FEL amplification process ($\phi_a$), the FEL amplification was kept reasonably low and only five (out of six) undulators were used. At these conditions, $\phi_a$ is supposed to be negligible. With these precautions, the main source of uncertainty in the GDD comes from the exact setting of the FEL and seeding parameters. According to our simulations, we can estimate the uncertainty on the GDD of the FEL to be ±100 fs$^2$.

At the end station, a pulsed valve was used at room temperature to create a pulsed beam of He atoms synchronized with the arrival of the XUV pulses. In the interaction region, the atomic beam intersected the laser pulses perpendicularly and the generated photoelectrons were detected with a magnetic bottle electron spectrometer. A retardation potential of 14 eV was applied to optimize the detector resolution. For the FEL settings used, the contribution of second harmonic FEL emission to the ionization yield is expected to be at least three orders of magnitude smaller and can thus be neglected. For the experimental parameters used, the space charge effects can be neglected as confirmed by measurements with different atom densities in the ionization volume. A distortion of photoelectron trajectories by the large retardation potentials was ruled out by simulations of the electron trajectories.

### Theory

To calculate the photoelectron spectra, we solve the time-dependent Schrödinger equation (TDSE) for a single-active-electron (SAE) model of the He atom. The effective potential in this model reads

$$V(r) = -\frac{1}{r}[1 + e^{-r/r_0} - re^{-r/r_1}], \tag{2}$$

where $r$ denotes the radial coordinate. It has the correct asymptotic behaviour for $r \to 0$ and $r \to \infty$ and the values of $r_0 = 0.5670$ Å and $r_1 = 0.4396$ Å guarantee that the binding energies $E_{1s^2} = -24.5874$ eV and $E_{1s2p} = -3.3694$ eV of He (ref. 54) are reproduced. The dipole moment in this model is a factor of 1.4 smaller than the NIST value[54], which is the

reason for the smaller AT splitting obtained in calculations compared with the experimental data. Field-free eigenstates up to angular momentum of $\ell = 3$ are calculated in a box of radius $R = 1.69 \times 10^4$ Å by means of the Numerov method and are used as a basis for the TDSE, which is solved in the velocity form. The box size $R$ is chosen sufficiently large to omit the need for absorbing boundary conditions. Thus, photoelectron spectra can be calculated directly from the occupations of the field-free eigenstates obtained in the propagation. Owing to the high intensities of interest, we treat the vector potential of the FEL pulse classically and use a Gaussian envelope. Thus, the vector potential reads

$$\mathbf{A}(t) = A_0\, g(t)\, \{\Re f(t), \Im f(t), 0\} \tag{3}$$

$$g(t) = \exp(-2\ln2\, t^2/T^2) \tag{3a}$$

$$f(t) = \exp(i(\omega_0 t + at^2)), \tag{3b}$$

where $A_0$ is the field amplitude, $T$ is the FEL pulse duration, which depends on the chirp, $\omega_0$ denotes the carrier frequency and $a$ is the linear chirp rate, which relates to the quadratic spectral phase coefficient $\phi_2$ (that is, GDD) as

$$a = \frac{\phi_2}{2\phi_2^2 + (T_0^2/\sqrt{8}\ln2)^2}, \tag{4}$$

where $T_0$ denotes the Fourier-transform-limited pulse duration.

To account for the experimental response function and the focal intensity averaging in the experiment, we calculated the average of the photoelectron spectra for a range of laser intensities (8.3 × 10$^1$–6.9 × 10$^{14}$ W cm$^{-2}$) and convoluted the result by the instrument response function (around 50 meV). In this way, the average intensity in the calculations is 2.74 × 10$^{14}$ W cm$^{-2}$, which matches the effective intensity in the experiment of $I_{eff}$ = 2.8 × 10$^{14}$ W cm$^{-2}$. These computationally intense simulations were performed for a few laser wavelengths and were used to calculate the data in Figs. 3b and 4c,d. We omitted a calculation of all laser wavelengths shown in Fig. 4b. Here, we show the calculations only for a single-intensity value equal to the effective intensity in the experiment.

### Data analysis

Background correction of the photoelectron spectra was done and the images were filtered for fluctuations in FEL pulse energy and photon energy. The effective intensity $I_{eff}$ was calibrated from the AT splitting taken at the maximum pulse energy according to

$$I_{eff} = 0.5\epsilon_0 c\left(\frac{\hbar\Omega}{\mu}\right)^2. \tag{5}$$

To this end, the AT splitting $\hbar\Omega$ was deduced by fitting the corresponding photoelectron spectrum with a sum of three Gaussian functions. For all other pulse energies, the prediction by equation (5) was plotted as dashed lines in Fig. 2a. The Rabi period was calculated based on the determined effective FEL intensity.

The low-intensity contribution in the data shown in Fig. 3a was removed by fitting the data with a sum of three Gaussians of which only the amplitude was fitted as the free parameter. The fitted Gaussian in the centre was subtracted from the data. For the data shown in Fig. 4a, the low-intensity distribution was removed by subtracting the photoelectron spectrum shown in Fig. 2c scaled in amplitude to account for the different pulse energies used in the two data sets.

To determine the ratio between the upper and lower photoelectron bands shown in Fig. 4c, we computed the integral of photoelectron intensity in the upper and lower bands for a photon energy of 21.22 eV (at $1s^2 \to 1s2p$ resonance) and divided the values.

## Data availability

Experimental data were generated at the FERMI large-scale facility. The experimental and simulation data included in this work are available on the open repository Zenodo at https://doi.org/10.5281/zenodo.14046711. Additional derived data supporting the findings of this study are available from the corresponding author upon reasonable request.

## Code availability

The code that supports the findings of the study is available from the corresponding authors upon reasonable request.

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

**Acknowledgements** We acknowledged the funding from the Bundesministerium für Bildung und Forschung (BMBF) LoKo-FEL (05K16VFB) and STAR (05K19VF3); the European Research Council (ERC) Starting Grant MULTIPLEX (101078689); the Deutsche Forschungsgemeinschaft (DFG) RTG 2717 and grant 429805582 (project SA 3470/4-1) and project STI 125/24-1; the Baden-Württemberg Stiftung Eliteprogram for Postdocs; the Swedish Research Council and Knut and Alice Wallenberg Foundation, Sweden; the Danish Agency for Science, Technology, and Innovation for funding through the instrument centre DanScatt. The research leading to this result has been supported by the COST Action CA21101 Confined Molecular Systems: From a New Generation of Materials to the Stars (COSY).

**Author contributions** L.B. conceived the experiment with input from U.S. and M.W.; E.A., M.D., A.D., I.N., F.P. and P.S. implemented and characterized the spectral phase shaping of the FEL. E.A., A.B., L.G., M. Manfredda, G.P., A.S., M.T. and M.Z. optimized the machine and the laser beam parameters. C.C., M.D.F. and O.P. managed the end station. F.R., B.A., G.C., K.D., S.D.G., N.G., S.H., F.L., Y.L., C.M., M. Michelbach, A.M., M. Mudrich, A.N., N.P., K.C.P., F.S., D.U., B.W., C.C, M.D.F., O.P. and L.B. performed the experiment with input from U.S., M.W., R.F., B.v.I., T.L., G.S. and R.J.S.; F.R. analysed the data under the supervision of L.B. and U.S. provided the theoretical calculations. L.B. wrote the paper with input from all authors.

**Competing interests** The authors declare no competing interests.

**Additional information**
**Correspondence and requests for materials** should be addressed to Lukas Bruder.

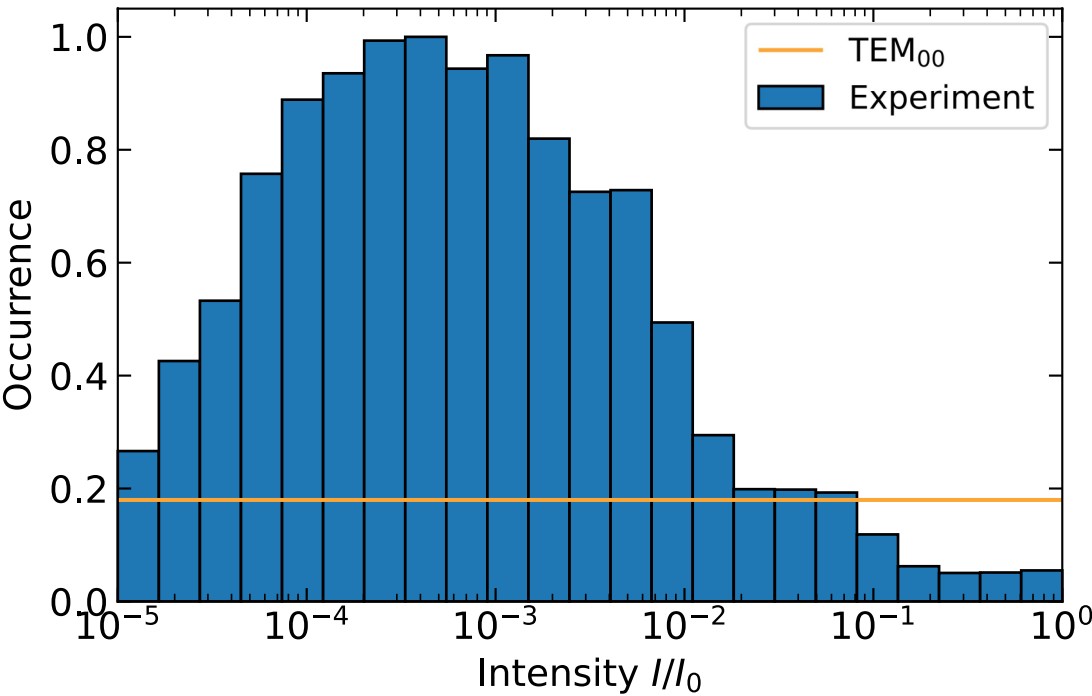

**Extended Data Fig. 1 | Spatial intensity distribution in the interaction volume of the experiment, measured with a Hartmann wavefront sensor.** The atomic jet target has a width of ≈ 0.2 mm along the FEL propagation direction, thus the intensity in the direction of propagation can be assumed to be constant. To visualize the intensity distribution in the transverse mode, we generated a histogram of the intensity values measured in the ionization volume (blue). The experimental distribution is compared to an ideal Gaussian $TEM_{00}$ mode (orange). While the $TEM_{00}$ mode is characterized by an equal relative occurrence of all intensity values in the ionization volume, the actual intensity occurrences measured in the experiment show a maximum at intensities roughly three orders of magnitude lower than the peak intensity $I_0$. Hence, in the experiment a much larger fraction of He atoms in the ionization volume were excited by lower intensities than expected theoretically. At these low intensities, the AT splitting is too small to be resolved. This rationalizes the appearance of a pronounced peak in the centre of the measured photoelectron spectra not showing an AT splitting.

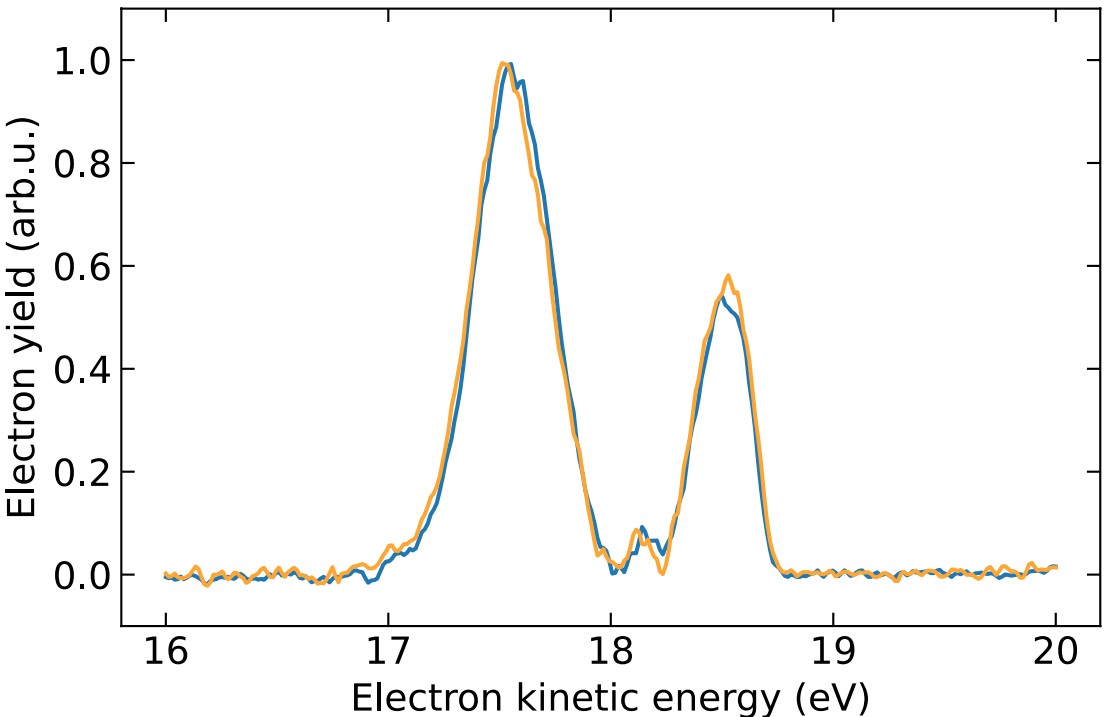

**Extended Data Fig. 2 | Demonstration of the reproducibility.** Examples of photoelectron spectra for GDD = 135 fs² taken before and after acquiring the data shown in Fig. 3a. The weak-field contribution has been removed in both spectra (see main text). Other than that no data processing is applied. Very good agreement between the two spectra is found, even though the chirp settings of the seed laser and thus of the FEL were changed in the range −1127 fs² to +695 fs² over the course of several hours between the two measurements.

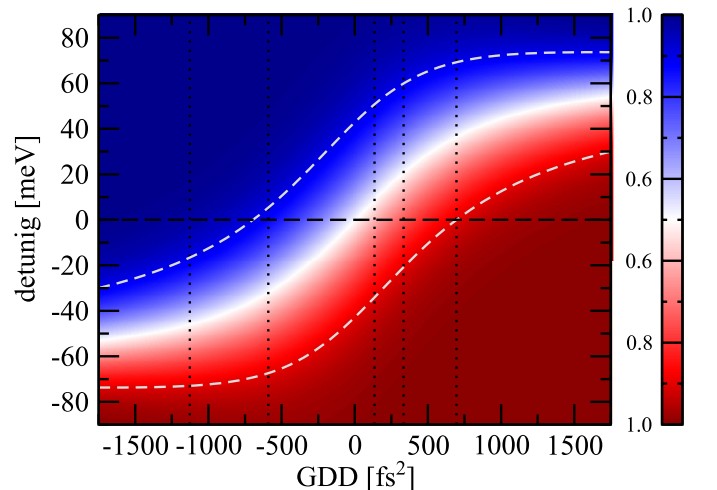

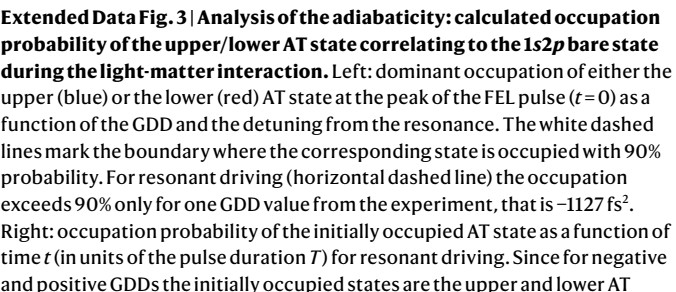

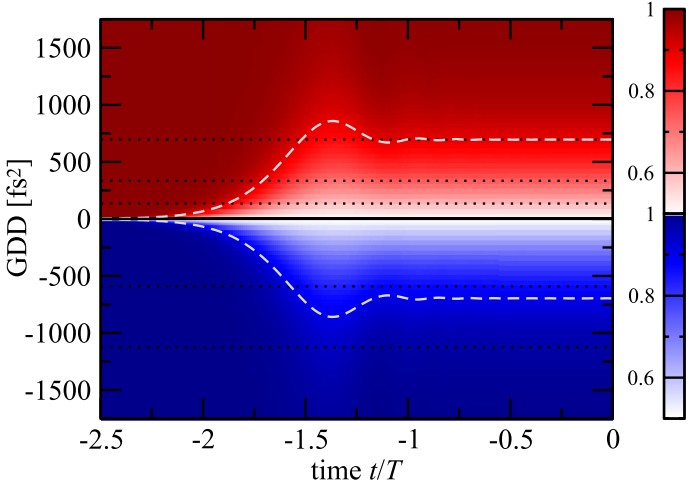

**Extended Data Fig. 3 | Analysis of the adiabaticity: calculated occupation probability of the upper/lower AT state correlating to the 1s2p bare state during the light-matter interaction.** Left: dominant occupation of either the upper (blue) or the lower (red) AT state at the peak of the FEL pulse ($t = 0$) as a function of the GDD and the detuning from the resonance. The white dashed lines mark the boundary where the corresponding state is occupied with 90% probability. For resonant driving (horizontal dashed line) the occupation exceeds 90% only for one GDD value from the experiment, that is −1127 fs². Right: occupation probability of the initially occupied AT state as a function of time $t$ (in units of the pulse duration $T$) for resonant driving. Since for negative and positive GDDs the initially occupied states are the upper and lower AT state, respectively, different colours are used. If the occupation remains ≥ 90% until the peak of the pulse ($t = 0$), which is the case for |GDD| ≥ 750 fs², the dynamic is adiabatic. As in the left panel, the dashed lines mark occupation of the initial state with 90% probability. In both panels the five experimental GDDs are marked with dotted lines. All calculations are done for a driven two-level system with the energy levels and the dipole coupling of helium for $T = 49.3$ fs and $I = 6 \times 10^{14}$ W/cm². The analysis reveals that the population transfer is only adiabatic for a frequency chirp with values of |GDD| ≥ 750 fs². Hence, the majority of the experiment is conducted in the non-adiabatic regime. The analysis also shows that the conditions for rapid adiabatic passage can be generally reached with the experimental approach.

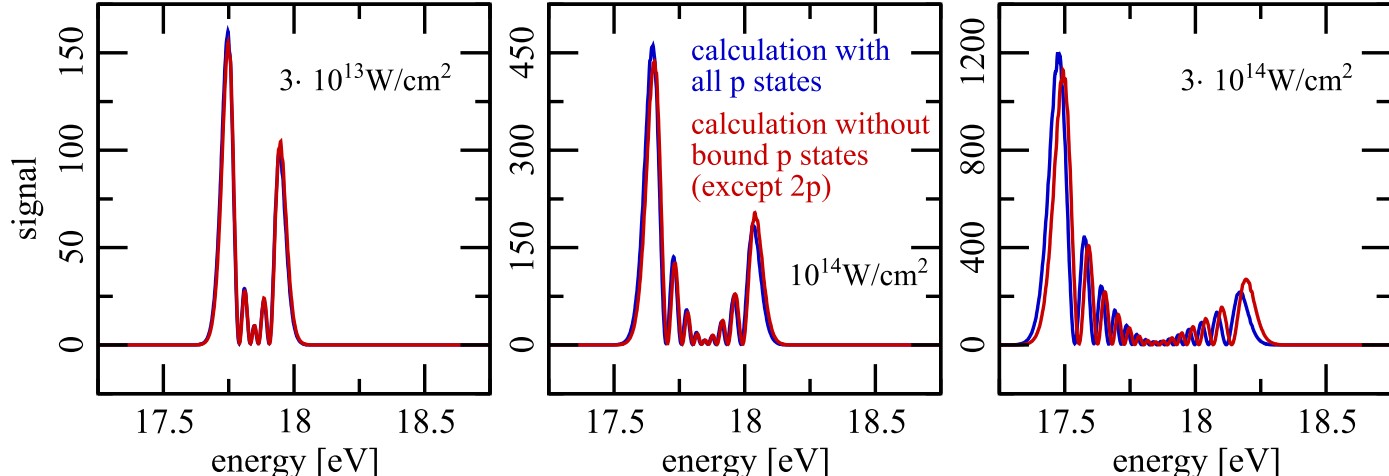

**Extended Data Fig. 4 | Influence of two-photon ionization via nearby states.** Calculated photoelectron spectra for resonant driving (pulse duration: 52.6 fs, GDD = 0 fs²) for three different peak intensities (as labelled). Blue: full TDSE-SAE model including all relevant He states. Red: suppressing two-photon ionization pathways via near-resonant states except for the 1s2p state. The latter case eliminates interference of multiple photoionization paths. The strong similarity between both photoelectron spectra confirms that photoionization paths via states energetically close to the 1s2p state play a negligible role. In particular, both spectra exhibit a clear asymmetry between the upper/lower AT states for a range of FEL intensities.