## [Peer Review File · Nature]

Strong-field quantum control in the extreme ultraviolet domain using pulse shaping

Corresponding Author: Dr Lukas Bruder

Version 0:

Reviewer comments:

Referee #1

(Remarks to the Author)

In this paper, the authors report on quantum control in Helium atoms using shaped extreme ultraviolet (XUV) pulses. A strong XUV field produced by a seeded free electron laser (FEL) dresses a resonant transition in Helium, leading to a large (~ 1 eV) Autler-Townes splitting of the $1s2p$ atomic energy levels. The authors show that shaping the FEL's seed laser leads to shaped XUV pulses from the FEL itself, with properties that mimic the spectral phase characteristics of the seed. This enables indirect pulse shaping in the XUV by far more straightforward pulse shaping of the near-infrared seed laser. In such a way, the authors show that the photoionization process in Helium can be controlled by chirping the frequency of the XUV pulses. Photoionization occurs preferentially via the higher energy or lower energy state of the Autler-Townes doublet, depending on the sign of the frequency chirp.

The XUV pulse shaping method is not a new technique, it was first reported in 2015 by Gauthier et. al. (Reference 52). Prior work in XUV quantum control mainly relied on XUV pulse sequences produced from FEL's, but I am not aware of experiments employing pulse shaping of the XUV pulses themselves, in the manner reported in Reference 52. This type of shaping is analogous to the well-established schemes of ultrafast pulse shaping in the visible and near-IR (reviewed in Reference 11). Adapting such schemes to the XUV could open up many exciting avenues for strong field control that have not yet been explored.

Despite the future potential of such experiments, I have a number of doubts about the methodology and interpretation of the results presented here. In my view the analysis does not yet justify the conclusions. Therefore this paper does not warrant publication in Nature in its current form.

My questions and comments for the authors:

1. The control mechanism, i.e. an adiabatic change in the carrier frequency of the FEL pulse with respect to time, is conceptually identical to established methods of rapid adiabatic passage (RAP). Realizations of RAP are discussed in References 50 and 51 and have been well understood for decades. Although prior realizations were done using visible and near-IR lasers, with detuned CW or shaped ultrafast pulses, the key physics presented in this paper is conceptually identical to most of the prior work on RAP. In moving to the XUV, the frequency of the driving laser and the ΔE (Autler-Townes energy splitting) are an order of magnitude larger, but the basic equations that describe the process (and therefore the physics) are essentially the same. Therefore the current manuscript does little to advance our understanding of RAP, or similar quantum control schemes.

2. There is no direct characterization of the FEL pulses except for a cross-correlation measurement at a single chirp value of the seed laser. As stated in the the Methods section, for the near transform-limited chirp value, the FEL pulse duration was measured via cross-correlation with the seed pulses. For the other chirp values, there is no evidence that the characterization of the FEL pulses is accurate, other than from the comparisons between the simulations and experiments (Figure 3). Cross-correlation does not extract the spectral phase of the XUV pulses. Experimental validation of the spectral phase in the XUV for each value of the seed laser chirp seems crucial for establishing the validity of the experimental method. In the paper by Gauthier et. al. (Reference 52), careful characterization of the shaped XUV pulses was performed for both positive and negative chirps. Were such characterizations done in these experiments?

3. The authors discuss the "aberrated intensity volume" in order to address the spatial dependence of the interaction. But what about the temporal dependence? In the same way that electrons are ionized at intensities lower than the peak (Figure 2 and Suppl. Fig. 1), presumably many electrons are also ionized well before the temporal peak of the pulse. The dressing and control process all occur within a single shaped XUV pulse, so temporal effects should be important, particularly as the pulse chirp is varied. What is the temporal dependence of the interaction?

4. In Figure 3, the comparison between theory and experiment raises a number of issues that are not addressed in the paper. The discussion in the paper mainly focuses on the lack of symmetry between the photoelectron (PE) distributions for positive and negative chirp. In other words, if the dynamics are purely due to RAP, with no other strong field effects playing a significant role, then one would observe a peak at the higher PE energy for negatively chirped pulses, and an identical peak (same PE magnitude and energy distribution) at the lower PE for positively chirped pulses with the same GDD (but with opposite sign).

Judging the results based on this assumption is problematic. Firstly, the measurements and calculations were not performed for the same positive and negative GDD values, so a direct comparison between chirps with opposite signs cannot be made. Secondly, the energy splitting between the dressed states is about 0.5 eV in theory and 1 eV in experiment. This seems like a fairly large discrepancy that is not addressed. Furthermore, the width of the PE distribution is much broader in the experimental data than in the simulations. The authors note that coupling of the dressed states to the continuum can lead to asymmetry in the PE distribution, but can such an effect account for the additional discrepancies between the experiments and the simulations?

5. The key claim of the paper -- realizing RAP with shaped ultrafast XUV pulses -- depends entirely on the assumption of adiabaticity in the interaction. However there is no discussion of adiabaticity, and no adiabaticity condition is given. Such conditions always boil down to a comparison of the couplings between the dressed states (driven by the laser/XUV field) and the energy level splittings between those states. What is the adiabaticity condition under these experimental conditions?

6. In Figure 4, the presentation of the plot makes it difficult to judge the degree of asymmetry for the negative chirps because their values approach zero. The non-perturbative simulations more closely matches the experimental data, so obviously the dynamics is dictated by a strong field effect. However, control schemes such as SPODS (Reference 50) apply to weak (perturbative) or intermediate (weakly non-perturbative) fields. Thus, the underlying control mechanism must be extended beyond SPODS, contrary to the simplified picture presented in Figure 1 and discussed on page 3 of the manuscript.

Strong field control schemes involving the continuum are not well understood at the moment, and experimental demonstrations of these effects are lacking. This leaves much room for questions about which are the dominant strong field effects that describe the interaction. The authors consider the influence of a chirp-dependent ionization rate but other effects are possible, for example, a Stark-like broadening mechanism that may affect the ionization dynamics.

Referee #2

(Remarks to the Author)

The authors report the demonstration of quantum control of dressed state dynamics using pulse shaping techniques in the XUV domain. This quantum control is achieved by applying the established spectral phase shaping techniques that exist for long wavelength lasers on the laser that is used to seed the FEL at FERMI. The paper also demonstrates coherent Rabi dynamics in helium at intensities more than 2.5×10^{14} W/cm², showing AC stark shifts of about 1 eV in the XUV regime. The results are somewhat similar to the results published in a recent earlier work by Nandi et al. (Ref. 26 in the manuscript), but at higher intensities. While the paper introduces intriguing new aspects, such as quantum control, its immediate application seems confined to the XUV regime, suggesting its suitability for a specialized readership.

Questions for authors:

- 1) The manuscript lacks clarity regarding the potential application of the results beyond the XUV region. Could the authors elaborate on the limitations of pulse shaping control? Specifically, are there constraints hindering the application of group delay dispersion (GDD) control to pulses with peak photon energies beyond ~20 eV? Similarly, are there restrictions with pulse duration?
- 2) The authors mention potential implications of pulse shaping techniques for X-rays several times. It would be valuable to assess the feasibility of such techniques with respect to current operational XFELs. If not feasible, this should be explicitly stated, considering factors such as the required pulse intensity and potential competing ionization processes.
- 3) It seems that a slightly different resonance transition was selected for this experiment compared to a similar previous experiment [26]. Was there a specific rationale behind choosing this transition, apart from the increased transition dipole moment? Considering that the other resonance (1s₂ → 1s_{4p}) is very close (within 2 eV) and represents an identical system, was there an effort made to replicate the results previously reported [26]?
- 4) Is the plot Fig2a for a flat phase ϕ ? Is the asymmetry present here between the photoelectron yield from the lower energy dressed state and the higher energy dressed state the same as the one referred to later at GDD = 0 fs² (Fig. 4c)? I realize that the uncertainty in the GDD may be large.
- 5) Figure 2a suggests a decrease in the energy bandwidth of the photoelectron distribution from left to right. Could the authors explain the reason for this?
- 6) In Fig. 3, How should one understand the emergence of a small peak for GDD values = 135 fs², -591 fs², -1127 fs² at

around $eKE = 18.1$ eV? If I understand correctly, any excitation due to focal averaging was subtracted out from these figures? Also, why does this peak not appear in the simulation?

7) Regarding the asymmetry in the AT doublet spectrum, simulations with photoionization interference channels turned off still result in asymmetry. It appears that this is the basis for saying the mechanism for the asymmetry in the AT doublet spectrum at play in Nandi et al.[26] is not manifesting here. The authors mention that "For helium such effects are, however, expected in a narrow parameter range [26, 58], which lies outside the regime probed in our experiment." Could the authors offer an intuitive explanation for why these interference effects are absent in this experiment?

Minor suggestions:

1) It would be useful if the authors briefly describe and provide an idea of group delay dispersion since it is the main parameter for quantum control. Perhaps specify the incident classical vector potential of the pulse with the GDD parameter in Sec.III. B.

Referee #3

(Remarks to the Author)

The manuscript by Richter et al. reports on a strong-field quantum control experiment achieved by pulse shaping in the EUV. The phase of the EUV pulses is achieved through intense laser seeding of the FEL. When the phase shaped pulse interacts with He atoms, the strong-field causes Autler-Townes (AT) energy splitting of an intermediate state (shown in Fig. 2). The population of the AT states is followed by the kinetic energy of the electrons following ionization. When the phase curvature is negative (positive) the lower (upper) state is preferentially populated (shown in Figs. 3 and 4).

I consider these results highly significant because they open the door to experiments with tailored spectral phases in the EUV. The presentation is clear, and the figures are high quality. The results support the claims in the abstract and the conclusions.

While the paper can be published as is, I suggest the authors consider the following:

Control of the curvature of the spectral phase opens the door for experiments along the lines of Silberberg and Dantus, where a sinusoidal function or a combination of second and third order spectral phase can be used to selectively excite two- and three-photon transitions. Similarly, spectral phase control has been used by Silberberg and Dantus to drive selectively stimulated-Raman transitions. These approaches are described in the following reviews [10.1146/annurev.physchem.040808.090427; 10.1002/cphc.200400342; Ref 32 in this manuscript, and references within these review articles] The advantage of these approaches to coherent control is that they are highly reproducible and have led to applications. In contrast, few if any experiments using genetic algorithms have been reproduced.

Version 1:

Reviewer comments:

Referee #1

(Remarks to the Author)

Review of Richter et al, "Strong field quantum control by pulse shaping in the extreme ultraviolet domain"

The revisions made by the authors provide a great deal of clarification that was lacking in the first version of the paper. The description of the experiment and results is more thorough and many important details have been sufficiently addressed.

In response to some of the the points made by the authors (authors comments in blue text):

We appreciate the reviewer's acknowledgement of the great perspectives and novelty of our work. We would like to clarify the important advancement in our work compared to Ref. 52. The work in Ref. 52 is constrained to the temporal compression of XUV pulses. In contrast, the coherent control of quantum dynamics in matter, as demonstrated in our work, involves many more degrees of freedom and is thus much more complex.

...

To make this point clearer, we have added the following statements to our manuscript:

p.2,§1: "...and chirp control for the temporal compression of XUV pulses [46] were recently demonstrated.."

p.3§4: "These demonstrations have been so far limited to applications of temporal compression and amplification of the FEL pulses. In contrast, the deterministic control of quantum dynamics in a material system involves many more degrees of freedom, which makes the situation considerably more complex."

I fully agree with this point. The ability to shape pulses was presented in Reference 52, but applying such shaped pulses to control quantum dynamics would represent a significant step. The sentences added by the authors clearly explain this advance.

1.1

We agree that the concept of RAP is well understood. However, the aim of our work is not adiabatic population transfer. Instead our aims are: (i) establishing the new concept of (strong-field) quantum control using phase-shaped XUV fields, (ii) revealing the strong-field induced dressing of continuum states, and (iii) demonstrating the effective control of the photoionization rate.

Based on this reply from the authors, and their reply to comment 1.3 below, crucial aspects of the experimental parameters and simulations are more transparent. In the experiment, the dynamics are adiabatic only at very large pulse chirps. A section that addresses the

adiabaticity of the measurement was added to the Supplementary Information. Similarly, SPODS reduces to RAP only in the limit of very large pulse chirps. This concept has also been clarified on p.3.

Furthermore, the authors highlight the importance of using XUV wavelengths to be able to access the multiphoton regime even at very high field strengths. This point was added to the main text on p.2.

The results in Figure 2 are a manifestation of SPODS, transferred into the XUV excitation regime. Regarding the temporal evolution of the dynamics (see also comment 1.3 below), they are described by the theory of SPODS, which treats the Rabi frequency analytically. If it is assumed that the physics here is the same as that for SPODS, which seems to be the case based the author's responses in comment 1.6 (below), then the TDSE simulation intrinsically captures the same ultrafast dynamics. This removes the need to dissect the temporal evolution of the process. In SPODS, the dressing step and the ionization step both occur within a single, shaped femtosecond pulse, which is also the case here. The key parameter though is the chirp parameter, which provides control over the quantum pathway. These dynamics are well reproduced by the experimental observable (electron KE), which fully connects the results of Figure 2 to SPODS.

However, my original concern was that certain results in the paper (Figures 1 and 2) were conceptually similar to existing, well-understood physics. The authors have borrowed from SPODS (a near-IR ultrafast quantum control technique) and shifted it into the XUV regime to create an XUV ultrafast quantum control technique. On its own, this would represent a important technical advance, but not an important physics advance.

To summarize these points, I think the three aims outlined above by the authors are well formulated. Figures 1 and 2 establish aim #1: Quantum control using shaped XUV fields. Aims #2 and #3 were not sufficiently established in the prior version of the paper, in my view. Thus, I was hesitant to recommend acceptance of the paper.

1.2

We indeed used the method described in Ref. 52 to confirm the validity of the chirp transfer from the seed laser pulses to the XUV pulses. This is stated in the methods section of the initially submitted manuscript (p.6 para. 2):

Thank you for highlighting this and adding more details about the XUV pulse characterization to the Supplementary Information.

1.3

In the experiment, the photoionization, and thus the probing of the dynamics, takes place during the whole light-matter interaction, as discussed in 3. Thus, the resulting photoelectron spectrum reflects the integral of the temporal dependence of the interaction. Resolving experimentally the temporal dependence in real-time would be very challenging. However, the main aspects of the dynamics, that are the ultrafast Rabi oscillations, the population transfer to the transient states and the coupling to the dressed continuum, are well reproduced in the experimental observable. Moreover, the theoretical model is based on the integration of the time-dependent Schrödinger equation, thus the temporal dependence of the interaction is fully and non-perturbatively included in the model. However, since the experimental observable reflects the temporal integral of the interaction, we refrain from a theoretical real-time analysis of the dynamics.

The previous version of the paper was vague as to the treatment of the XUV field in the TDSE calculations. Important details have now been added to Section 3 B (Methods).

1.4

The discrepancies between experiment and theory w.r.t the shape and width of the photoelectron distributions are due to the intensity average in the laser focus of the experiment. In our initial submission we decided against including this experimental broadening mechanism in the model to avoid the blurring of details. We agree that this can be misleading and have now included this effect in our model (see Fig. 3 in revised manuscript).

Correcting Figure 3 to include the effect of broadening is an important change and allows the experiment and simulations to be compared much more clearly. Figure R2 is also very helpful, although I am still not sure why the authors did not include at least one such case (chirps of equal magnitudes and opposite signs) in Figure 3.

Figures R2 and the revised Figure 3 are now much more convincing in establishing Aim #2: the strong field dressing of continuum states.

1.5

We regret that the key claim of our work was not expressed more clearly in our initial submission. While RAP in the XUV and soft X-ray domain would be an appealing application of our work, this is not the focus of our current work (see comment 1.1).

In light of the authors' earlier comment (1.1) and my comments above, I accept their clarification.

1.6

However, due to the large intensities in our study, the coupling to the continuum is modified, which goes indeed beyond conventional SPODS experiments where only the bound state dynamics were controlled. This difference can be seen from the measured asymmetry (in the new manuscript version already discussed in Fig.1) and the chirp-dependent ionization rate. In our fully non-perturbative model (described in detail in Ref. 36 of initial submission), both effects can be traced back to the strong-field modified coupling to the continuum. Hence, we provide here a detailed analysis for the extension of conventional SPODS.

I think the author's arguments are convincing and help establish Aim #3: demonstrating the effective control of the ionization rate through strong field quantum control. The four additions and corrections (to Figure 1 and text on p.2, p.3, and p.5) are important for understanding the author's claims.

In conclusion, the revised manuscript presents the key physics (based around Figures 3 and 4) in a much stronger way. The interpretation of these results is more convincing. XUV pulse shaping in general holds great promise for opening up a new paradigm in strong field quantum control. I would now recommend the manuscript for publication in Nature.

Referee #2

(Remarks to the Author)

The authors have thoroughly addressed my comments in their response and have made the necessary revisions to the manuscript. I am satisfied with the changes and improvements made, which significantly enhance the overall quality of the work. The work opens the door for exciting advancements in quantum control with XUV light. I now recommend the publication of the manuscript.

Answers to referee comments, manuscript # 2024-02-03782

Dear Editor,

We appreciate the invitation for re-submission of our manuscript and we thank the reviewers for their valuable comments and their clear support of our work. Please find below our response to the reviewer comments and other changes made in the manuscript (highlighted in blue).

Response to referee #1 :

In this paper, the authors report on quantum control in Helium atoms using shaped extreme ultraviolet (XUV) pulses. A strong XUV field produced by a seeded free electron laser (FEL) dresses a resonant transition in Helium, leading to a large (~ 1 eV) Autler-Townes splitting of the $1s2p$ atomic energy levels. The authors show that shaping the FEL's seed laser leads to shaped XUV pulses from the FEL itself, with properties that mimic the spectral phase characteristics of the seed. This enables indirect pulse shaping in the XUV by far more straightforward pulse shaping of the near-infrared seed laser. In such a way, the authors show that the photoionization process in Helium can be controlled by chirping the frequency of the XUV pulses. Photoionization occurs preferentially via the higher energy or lower energy state of the Autler-Townes doublet, depending on the sign of the frequency chirp.

The XUV pulse shaping method is not a new technique, it was first reported in 2015 by Gauthier et al. (Reference 52). Prior work in XUV quantum control mainly relied on XUV pulse sequences produced from FEL's, but I am not aware of experiments employing pulse shaping of the XUV pulses themselves, in the manner reported in Reference 52. This type of shaping is analogous to the well-established schemes of ultrafast pulse shaping in the visible and near-IR (reviewed in Reference 11). Adapting such schemes to the XUV could open up many exciting avenues for strong field control that have not yet been explored.

Despite the future potential of such experiments, I have a number of doubts about the methodology and interpretation of the results presented here. In my view the analysis does not yet justify the conclusions. Therefore this paper does not warrant publication in Nature in its current form.

We appreciate the reviewer's acknowledgement of the great perspectives and novelty of our work. We would like to clarify the important advancement in our work compared to Ref. 52. The work in Ref. 52 is constrained to the temporal compression of XUV pulses. In contrast, the coherent control of quantum dynamics in matter, as demonstrated in our work, involves many more degrees of freedom and is thus much more complex. In particular, quantitative uncertainties of the pulse shaping method were not discussed in Ref. 52. Hence, it was not clear if applications like deterministic coherent control are possible. Our work shows for the first time that such applications are indeed feasible with high fidelity and is therefore expected to open up a whole new stream of research and serve as a reference for future developments in this field.

To make this point clearer, we have added the following statements to our manuscript:

p.2,§1: "...and chirp control for the temporal compression of XUV pulses [46] were recently demonstrated.."

p.3§4: "These demonstrations have been so far limited to applications of temporal compression and amplification of the FEL pulses. In contrast, the deterministic control of quantum dynamics in a

material system involves many more degrees of freedom, which makes the situation considerably more complex.”

My questions and comments for the authors:

1.1

The control mechanism, i.e. an adiabatic change in the carrier frequency of the FEL pulse with respect to time, is conceptually identical to established methods of rapid adiabatic passage (RAP). Realizations of RAP are discussed in References 50 and 51 and have been well understood for decades. Although prior realizations were done using visible and near-IR lasers, with detuned CW or shaped ultrafast pulses, the key physics presented in this paper is conceptually identical to most of the prior work on RAP. In moving to the XUV, the frequency of the driving laser and the ΔE (Autler-Townes energy splitting) are an order of magnitude larger, but the basic equations that describe the process (and therefore the physics) are essentially the same. Therefore the current manuscript does little to advance our understanding of RAP, or similar quantum control schemes.

We agree that the concept of RAP is well understood. However, the aim of our work is not adiabatic population transfer. Instead our aims are: (i) establishing the new concept of (strong-field) quantum control using phase-shaped XUV fields, (ii) revealing the strong-field induced dressing of continuum states, and (iii) demonstrating the effective control of the photoionization rate.

For the demonstration of (i) we intentionally use a known process which is crucial for a precise and reliable assessment of our new approach. With (ii) we show, that our work provides indeed new insight into strong-field phenomena beyond experiments performed at longer wavelengths. The high field strength applied in our study corresponds to a Keldysh parameter of 11 (multiphoton ionization regime). In contrast, applying the same field strength in an analogous NIR experiment, e.g. Ref. 51, would correspond to a Keldysh parameter of 0.35, thus, the quantum dynamics would be dominated by tunnel and above barrier ionization. Hence, at short wavelengths a very different regime becomes accessible: field-strengths sufficient to induce a mixing of continuum states can be applied, while at the same time the dynamics of the target system (bound atomic states) are still in the multiphoton ionization regime. This unique scenario enables the sensing of the dressed continuum using the well-understood quantum dynamics of the atomic system. Moreover, in (iii) we demonstrate how this interplay between dressed bound and continuum states can be exploited to control the total ionization rate and to achieve the trapping of populations in the bound excited state. To our knowledge, this mechanism is not attainable at longer wavelengths.

These examples show that the strong-field physics in our study is indeed very different from previous NIR/VIS experiments and show how new aspects become accessible by strong-field quantum control at short wavelengths. In particular, reviewer #1 points out, that "Strong field control schemes involving the continuum are not well understood at the moment, and experimental demonstrations of these effects are lacking." (comment 1.6), which also supports the novelty of our work.

We further stress that the control mechanism applied in our study is different from RAP. We probe and control the populations of transient states induced during the light-matter interaction and we control the overall photoionization rate, while in RAP the focus lies on the population transfer to a final (bound) target state reached after the light-matter interaction. In particular, our mechanism does not require adiabaticity, as discussed in Ref. 50. To make this difference clearer, we calculated the adiabaticity condition for our experiment, as suggested by the reviewer (see comment 1.5). The calculation reveals, that the majority of our experiment is in the non-adiabatic regime. However, we agree with the reviewer that RAP would be an appealing application of our approach. While RAP has

not been demonstrated at short wavelengths to date, our work establishes for the first time the necessary prerequisites for such applications.

To make these points clearer to the reader, we made the following changes in our manuscript:

p1, abstract: “Our approach unravels a strong dressing of the ionization continuum, otherwise elusive to experimental observables. The latter is exploited to achieve control of the total ionization rate...”

p2§2: “The high radiation intensity also causes a strong dressing of the photoelectron continuum, while the ionization dynamics of the atomic system are still in the multiphoton regime (Keldysh parameter $\gamma = 11$). In contrast, the dynamics of a system dressed with NIR radiation of comparable intensity, would be dominated by tunnel and above barrier ionization ($\gamma = 0.35$) [8]. Hence, the use of short-wavelength radiation provides access to a unique regime, where the interplay between strongly dressed bound states and a strongly dressed continuum can be studied”

P3§3: “Here, we extend SPODS to the XUV domain and include a new physical aspect, that is the transition of the bound atomic system into a strongly-dressed continuum.”

P4§2: “However, this also shows that the condition for rapid adiabatic passage [2] can be generally reached with our approach, unlocking new possibilities for efficient population transfer in the XUV and potentially in the soft X-ray regime.”

1.2

There is no direct characterization of the FEL pulses except for a cross-correlation measurement at a single chirp value of the seed laser. As stated in the the Methods section, for the near transform-limited chirp value, the FEL pulse duration was measured via cross-correlation with the seed pulses. For the other chirp values, there is no evidence that the characterization of the FEL pulses is accurate, other than from the comparisons between the simulations and experiments (Figure 3). Cross-correlation does not extract the spectral phase of the XUV pulses. Experimental validation of the spectral phase in the XUV for each value of the seed laser chirp seems crucial for establishing the validity of the experimental method. In the paper by Gauthier et. al. (Reference 52), careful characterization of the shaped XUV pulses was performed for both positive and negative chirps. Were such characterizations done in these experiments?

We indeed used the method described in Ref. 52 to confirm the validity of the chirp transfer from the seed laser pulses to the XUV pulses. This is stated in the methods section of the initially submitted manuscript (p.6 para. 2):

“The coherent transfer of the seed phase ϕ_s to the FEL phase ϕ_{nH} was characterized for a set of seed laser and FEL settings prior to the beamtime using a procedure outlined in Ref. [52]. To minimize the additional chirp introduced by the FEL amplification process (ϕ_a), the FEL amplification was kept reasonably low and only five (out of six) undulators were used. At these conditions, ϕ_a is supposed to be negligible. With these precautions and based on the seed laser GDD, we estimate the uncertainty on the GDD of the FEL to be $\pm 100 \text{ fs}^2$.”

In general, the characterization of XUV pulses remains an experimental challenge. There is hardly any method available for reliable characterization of the spectral phase of XUV pulses. However, for seeded FELs, we can simulate the full high-gain harmonic generation (HG) process in the FEL with

high detail using the GENESIS 1.3 code¹. This code is capable of simulating the FEL process for chirped seed laser pulses and was already tested in several FEL experiments. The details of these numerically intense simulations are, however, beyond the scope of this manuscript and will be published in a general accelerator physics context in the near future.

Figure R1: Validation of the GENESIS simulations for spectral phase transfer from the seed laser to the FEL output. The method from Ref. 52 is used. Labels indicate the predicted GDD values from the FEL simulation. For these values excellent agreement between experiment and theory is found. Taken from Ref. 2.

As mentioned above, we used the method described in Ref. 52 for additional experimental validation of the simulations in a preparation campaign prior to the beamtime, using the same experimental setup as during the beamtime. Confirmation was done for several GDD values covering a large range (+200 fs², -2500 fs², -3800 fs²). These characterizations are published in a separate paper² (see also figure R1). We thus do not show this data in the current manuscript but rather refer to the published paper.

Furthermore, the FEL simulations have shown that for a given laser chirp, the GDD of the FEL pulse can vary by up to 100fs², depending on the specific FEL and seeding settings. For this reason, we assumed an uncertainty in the FEL GDD of $\pm 100\text{fs}^2$.

To make this point clearer to the reader, we modified the statements in the Methods section:

P6§3: “The coherent transfer of the seed phase ϕ_s to the FEL phase ϕ_{nH} was with FEL simulations using the GENESIS 1.3 code [65] and a FEL model [66]. With these tools, the FEL was analyzed as outlined in Ref. [46] for a set of seed laser and FEL settings prior to the beamtime. Details can be found in Ref. [66]. ... With these precautions the major source of uncertainty in the GDD comes from the exact setting of the FEL and seeding parameters. According to our simulations we can estimate the uncertainty on the GDD of the FEL to be $\pm 100\text{fs}^2$.”

¹ S. Reiche, GENESIS 1.3: A Fully 3D Time-Dependent FEL Simulation Code, Nuclear Instruments and Methods in Physics Research Section A: Accelerators, Spectrometers, Detectors and Associated Equipment 429, 243 (1999).

² F. Pannek et al., 14th International Particle Accelerator Conference, JACoW Publishing, Geneva, Switzerland, Italy, 2023), pp. 1954–1957.

1.3

The authors discuss the "aberrated intensity volume" in order to address the spatial dependence of the interaction. But what about the temporal dependence? In the same way that electrons are ionized at intensities lower than the peak (Figure 2 and Suppl. Fig. 1), presumably many electrons are also ionized well before the temporal peak of the pulse. The dressing and control process all occur within a single shaped XUV pulse, so temporal effects should be important, particularly as the pulse chirp is varied. What is the temporal dependence of the interaction?

In the experiment, the photoionization, and thus the probing of the dynamics, takes place during the whole light-matter interaction, as discussed in ³. Thus, the resulting photoelectron spectrum reflects the integral of the temporal dependence of the interaction. Resolving experimentally the temporal dependence in real-time would be very challenging. However, the main aspects of the dynamics, that are the ultrafast Rabi oscillations, the population transfer to the transient states and the coupling to the dressed continuum, are well reproduced in the experimental observable. Moreover, the theoretical model is based on the integration of the time-dependent Schrödinger equation, thus the temporal dependence of the interaction is fully and non-perturbatively included in the model. However, since the experimental observable reflects the temporal integral of the interaction, we refrain from a theoretical real-time analysis of the dynamics.

To clarify this point, we made the following modification in our manuscript:

P2§3: "This is achieved by immediate photoionization over the course of the femtosecond pulses, thus projecting the time-integrated energy level shifts onto the electron kinetic energy (eKE) distribution (Fig. 1b)."

1.4

In Figure 3, the comparison between theory and experiment raises a number of issues that are not addressed in the paper. The discussion in the paper mainly focuses on the lack of symmetry between the photoelectron (PE) distributions for positive and negative chirp. In other words, if the dynamics are purely due to RAP, with no other strong field effects playing a significant role, then one would observe a peak at the higher PE energy for negatively chirped pulses, and an identical peak (same PE magnitude and energy distribution) at the lower PE for positively chirped pulses with the same GDD (but with opposite sign).

Judging the results based on this assumption is problematic. Firstly, the measurements and calculations were not performed for the same positive and negative GDD values, so a direct comparison between chirps with opposite signs cannot be made. Secondly, the energy splitting between the dressed states is about 0.5 eV in theory and 1 eV in experiment. This seems like a fairly large discrepancy that is not addressed. Furthermore, the width of the PE distribution is much broader in the experimental data than in the simulations. The authors note that coupling of the dressed states to the continuum can lead to asymmetry in the PE distribution, but can such an effect account for the additional discrepancies between the experiments and the simulations?

The discrepancies between experiment and theory w.r.t the shape and width of the photoelectron distributions are due to the intensity average in the laser focus of the experiment. In our initial submission we decided against including this experimental broadening mechanism in the model to

³ M. Wollenhaupt, A. Assion, O. Bazhan, Ch. Horn, D. Liese, Ch. Sarpe-Tudoran, M. Winter, and T. Baumert, Control of Interferences in an Autler-Townes Doublet: Symmetry of Control Parameters, Phys. Rev. A 68, (2003).

avoid the blurring of details. We agree that this can be misleading and have now included this effect in our model (see Fig. 3 in revised manuscript). Moreover, the model underestimates the transition dipole moment of He by a factor of 1.4. Taking this into account, the AT splitting in the experiment (1.02 eV) and theory (0.74 eV) are in good agreement. Constraining in the model the dipole moment to the literature value results in unrealistic shapes of the photoelectron spectra and was therefore forgone. In overall, by including these effects into the simulated photoelectron spectra, the main features of the experiment are well reproduced.

We agree with the reviewer, that one approach to extract information about the coupling to the continuum would be the comparison of the photoelectron yields for one large but identical positive and negative chirp value (large enough to be in the adiabatic regime, see also comment 1.5). For this case the photoelectron spectra should be perfectly symmetric and any asymmetry would imply additional effects such as the coupling to the dressed continuum. Instead, in Fig. 4c we extract the information about the continuum dressing by including all five measured chirp values, which increases the statistics and includes information about the general chirp dependence of the effect. We thus believe the information content in our approach is higher. To account for the reviewer's comment we show below (Fig. R2) the theoretical calculation for symmetric chirp values in the adiabatic regime of the experiment (see also comment 1.5). A clear asymmetry can be observed, which is assigned to the coupling to the dressed continuum.

Figure R2: Comparison of calculated photoelectron spectra for identical positive/negative chirp values for driving the $1s^2 \rightarrow 1s2p$ transition on resonance in the adiabatic regime. Single laser intensity of $I=2.8 \times 10^{14} \text{ W/cm}^2$ (dark color), intensity averaged data for $I = 8.3 \times 10^{13} - 6.9 \times 10^{14} \text{ W/cm}^2$ (light color).

In addition, we made the following changes to the manuscript:

P4§3: “To account for experimental broadening effects, we calculated the photoelectron spectra for a single intensity (corresponding to the experimental I_{eff}) and including the focal intensity average present in the experiment (see Methods for details). All salient features of the experiment are well reproduced: The control of the dressed state populations is in very good qualitative agreement. The different widths and shapes of the photoelectron peaks are qualitatively well matched between experiment and the calculations. The difference in the AT energy splitting between experiment ($\Delta E_{\text{exp}} \approx 1.02 \text{ eV}$) and theory ($\Delta E_{\text{theo}} = 0.74 \text{ eV}$) is in good agreement with the fact, that the model underestimates the transition dipole moment of the $1s^2 \rightarrow 1s2p$ transition by a factor of 1.41 (see Methods section).”

p.7§2: “To account for the experimental response function and the focal intensity averaging in the experiment, we calculated the average of the photoelectron spectra for a range of laser intensities ($8.3 \times 10^{13} - 6.9 \times 10^{14} \text{ W/cm}^2$) and convoluted the result by the instrument response function ($\approx 50 \text{ meV}$). In this way, the average intensity in the calculations is $2.74 \times 10^{14} \text{ W/cm}^2$, which matches the effective intensity in the experiment of $I_{\text{eff}} = 2.8 \times 10^{14} \text{ W/cm}^2$. These computationally intense simulations were performed for a few laser wavelengths and were used to calculate the data in Fig. 3b, 4c,d. We omitted a calculation of all laser wavelengths disclosed in Fig. 4b. Here, we show the calculations only for a single intensity value equal to the effective intensity in the experiment.”

1.5

The key claim of the paper -- realizing RAP with shaped ultrafast XUV pulses -- depends entirely on the assumption of adiabaticity in the interaction. However there is no discussion of adiabaticity, and no adiabaticity condition is given. Such conditions always boil down to a comparison of the couplings between the dressed states (driven by the laser/XUV field) and the energy level splittings between those states. What is the adiabaticity condition under these experimental conditions?

We regret that the key claim of our work was not expressed more clearly in our initial submission. While RAP in the XUV and soft X-ray domain would be an appealing application of our work, this is not the focus of our current work (see comment 1.1). We agree with the reviewer that a discussion of the adiabaticity condition will help to avoid the confusion between our work and RAP. To analyze the adiabaticity condition, we make use of our theory model covering the full non-perturbative dynamics, based on which we calculated directly the coupling between the adiabatic states. This provides a direct and robust criterium of adiabaticity. We added this analysis to the Supp. Info. (see Supp. Info III). The analysis reveals, that the majority of the experiment is conducted in the non-adiabatic regime. This agrees with the fact that the applied quantum control concept does not rely on adiabaticity, as also discussed in Ref. 50.

In addition to the data added to the Supp. Info, we added the following explanation to the main text:

P4§3: “We note, that the implemented control scheme is not restricted to adiabatic processes [51]. In fact, in the presented experiment the dynamics are only adiabatic for the largest frequency chirp (which corresponds to $\text{GDD} = -1127 \text{ fs}^2$) (see Supp. Info. III).”

1.6

In Figure 4, the presentation of the plot makes it difficult to judge the degree of asymmetry for the negative chirps because their values approach zero. The non-perturbative simulations more closely matches the experimental data, so obviously the dynamics is dictated by a strong field effect. However, control schemes such as SPODS (Reference 50) apply to weak (perturbative) or intermediate) (weakly non-perturbative) fields. Thus, the underlying control mechanism must be extended beyond SPODS, contrary to the simplified picture presented in Figure 1 and discussed on page 3 of the manuscript.

Strong field control schemes involving the continuum are not well understood at the moment, and experimental demonstrations of these effects are lacking. This leaves much room for questions about which are the dominant strong field effects that describe the interaction. The authors consider the influence of a chirp-dependent ionization rate but other effects are possible, for example, a Stark-like broadening mechanism that may affect the ionization dynamics.

We appreciate the reviewer's acknowledgement of the novelty of our work by mentioning that „Strong field control schemes involving the continuum are not well understood at the moment, and experimental demonstrations of these effects are lacking“. In our work, we exactly provide such an experimental demonstration by combining SPODS with the coupling to a strongly dressed continuum. SPODS itself is well understood as also acknowledged by the reviewer (comment 1.1: „...discussed in References 50 and 51 and have been well understood for decades.“). Previous experiments and theoretical analysis of SPODS were conducted in the multiphoton ionization regime at intensities where multiple Rabi cycles are driven. The same conditions apply in our experiments and thus are not expected to exceed the validity of SPODS, and the known description of SPODS can be used. Likewise, since our experiment is conducted in the multiphoton ionization regime, other strong-field processes such as tunnel and above barrier ionization can be neglected (see also our answer to comment 1.1).

However, due to the large intensities in our study, the coupling to the continuum is modified, which goes indeed beyond conventional SPODS experiments where only the bound state dynamics were controlled. This difference can be seen from the measured asymmetry (in the new manuscript version already discussed in Fig.1) and the chirp-dependent ionization rate. In our fully non-perturbative model (described in detail in Ref. 36 of initial submission), both effects can be traced back to the strong-field modified coupling to the continuum. Hence, we provide here a detailed analysis for the extension of conventional SPODS.

As an additional strong-field effect, there is a ponderomotive shift (which is the most prominent strong-field modification of the continuum). It is in our study less than 100 meV and, hence, much smaller than the AT splitting. This splitting is by far the most important Stark-like (dynamic Stark shift) effect as can be seen in the photoelectron spectra. The low atom density ($< 10^{10}/\text{cm}^3$) in the experiment indicates that Stark-broadening, as it occurs in plasmas, can be ruled out. To account for the reviewer's comment we made the following changes to manuscript:

We changed the representation in Fig. 1a-c.

P2§4: “In analogy to the bound state description, the dressed continuum states are obtained by a diagonalization of the corresponding Hamiltonian. The hybrid electron-photon eigenstates compose of a mixing of partial waves with different angular momenta, which alters the coupling strength to the dressed bound states of the He atoms (Fig. 1a).”

P3§3: “which is well established in the NIR spectral domain [51]. Here, we extend SPODS to the XUV domain and include a new physical aspect, that is the coupling of the bound system to a strongly-dressed ionization continuum.”

P5§2: “Moreover, due to the large Keldysh parameter ($\gamma = 11$) and the low ponderomotive potential ($U_p < 100$ meV) in our study, other strong-field effects are expected to play a negligible role in the observed dynamics.”

Response to referee #2 :

The authors report the demonstration of quantum control of dressed state dynamics using pulse shaping techniques in the XUV domain. This quantum control is achieved by applying the established spectral phase shaping techniques that exist for long wavelength lasers on the laser that is used to seed the FEL at FERMI. The paper also demonstrates coherent Rabi dynamics in helium at intensities more than 2.5×10^{14} W/cm², showing AC stark shifts of about 1 eV in the XUV regime. The results are somewhat similar to the results published in a recent earlier work by Nandi et al. (Ref. 26 in the

manuscript), but at higher intensities. While the paper introduces intriguing new aspects, such as quantum control, its immediate application seems confined to the XUV regime, suggesting its suitability for a specialized readership.

We would like to clarify the novelty of our work w.r.t. the work by Nandi et al. (Ref. 26). Ref. 26 is an important demonstration of strong-field induced dynamics driven by XUV light. The main achievement in Ref. 26 is the observation of Rabi dynamics in He atoms and their theoretical description. In contrast, our achievements are very different: (i) establishing the new concept of (strong-field) quantum control using phase-shaped XUV fields, (ii) revealing the strong-field induced dressing of continuum states and (iii) the effective control of the photoionization rate (see also comment 1.1). None of these aspects are reported in Ref. 26. The novelty and impact of our achievements are pointed out by all reviewers: “Adapting such schemes to the XUV could open up many exciting avenues for strong field control that have not yet been explored.” (reviewer #1), “the paper introduces intriguing new aspects, such as quantum control” (reviewer #2), and “I consider these results highly significant because they open the door to experiments with tailored spectral phases in the EUV.” (reviewer #3).

The only similarity of our work with Ref. 26 is the fact that we also observe Rabi dynamics in He atoms. However, we only use the Rabi cycling as a tool to demonstrate the above-mentioned achievements. Hence, we are greatly extending on the work of Ref. 26. In addition, the insights into strong-field quantum dynamics gained in our work are very different from the conclusions drawn in Nandi et al. (see comment 2.7). We therefore believe that our work is indeed very distinct from that reported in Ref. 26. Further details and corresponding changes to the manuscript can be found in the specific answers to the comments below.

Questions for authors:

2.1

The manuscript lacks clarity regarding the potential application of the results beyond the XUV region. Could the authors elaborate on the limitations of pulse shaping control? Specifically, are there constraints hindering the application of group delay dispersion (GDD) control to pulses with peak photon energies beyond ~ 20 eV? Similarly, are there restrictions with pulse duration?

In terms of attainable photon energies, an asset of seeded FELs over other coherent XUV light sources, is the facile upscaling of photon energies. The employed high-gain harmonic generation (HG) principle works reliably and robustly down to wavelengths of ≈ 20 nm⁴, which implies the feasibility of spectro-temporal pulse shaping down to these wavelengths. This enables already a number of highly interesting, so far unattained applications. E.g. coherent control with atomic-selectivity in halogenated molecules by addressing localized iodine core electrons, or the investigation and quantum control of electron correlations in fundamental quantum systems, for instance, in doubly-excited helium.

The next step in the evolution of seeded FELs is the echo-enabled harmonic generation (EEHG). Recent first demonstrations showed a realistic perspective to extend the attainable photon energy to 620 eV⁵, covering the K-edges of C, N and O. Specifically, the compatibility of EEHG with spectro-temporal pulse shaping has been shown experimentally⁶, verifying the feasibility of our coherent

⁴ E. Allaria et al. Nat Photon 6, 10 (2012).

⁵ P. Rebernik Ribič et al., Coherent Soft X-Ray Pulses from an Echo-Enabled Harmonic Generation Free-Electron Laser, Nat Photon 13, 8 (2019).

⁶ N. S. Mirian et al., Phys. Rev. Accel. Beams 23, 060701 (2020).

control approach using EEHG. These initial demonstrations of EEHG have proven that scaling the photon energy to the soft X-ray regime is only a matter of technical upgrades of the machine. Besides, the seeded operation of FELs implicitly requires a coherent phase transfer from the seed laser to the FEL output⁷. Hence, any demonstration of seeded operation implies that also our pulse shaping approach is conceptually feasible. Thus, an extension of the pulse shaping concept to the soft X-ray regime is within reach.

We note, that the development of EEHG is very dynamic. Besides activities at FERMI, the agenda of several other FEL facilities (FLASH, SwissFEL, Shanghai SXFEL) include EEHG for operation in the soft X-ray domain as a high priority⁸. Thus, the new perspectives unfolded by our work will impact and foster developments beyond the FERMI facility, thereby influencing a much larger community.

In terms of pulse shaping capabilities, the constraints are defined by the minimum requirements for seeded FEL operation: (i) minimum seed laser peak power above a given threshold (100 - 250 MW, depending on photon energy) and (ii) seed pulse duration limited by the duration of the e-beam (≤ 1 ps) and by the bandwidth of the FEL harmonic conversion and amplification process (≥ 50 fs, also depending on photon energy). The duration of the resulting FEL pulses at the n 'th harmonic scales with the input seed pulse duration and with the harmonic order as $1/\sqrt{n}$ ⁹. Hence, pulses with duration ranging from sub-15 fs to a few hundreds of femtoseconds can be realistically generated and shaped with current HGHG technology and will extend to the few-fs regime with EEHG technology.

In terms of attainable tailored pulse shapes, the high intensity provided by ultrafast lasers that are currently available as seed sources offers a wide window of spectro-temporal shaping satisfying conditions (i) and (ii). For instance, seed laser pulses with energy of 300 μ J can be stretched to 1 ps (15000 fs²) to fulfill these requirements. Indeed, FEL operation with seed laser pulses chirped by $GDD=7400$ fs² was demonstrated¹⁰. Even attosecond pulse trains can be generated¹¹ and may be phase shaped with this approach, opening-up further, unprecedented possibilities.

In summary, the listed experiments show a clear perspective for the extension of the pulse shaping technique w.r.t. the temporal and photon energy range. While theoretical models would be desirable to further extrapolate the extension of our approach, such simulations strongly depend on the noise conditions in the experimental setup which cannot be predicted reliably without performing the actual experiments. We therefore refrain from such model predictions. However, we note that from the basic operation principle of seeded FELs there is no fundamental limit of our pulse shaping approach.

To make the technical feasibility of potential future applications clearer, we added the following statements to the manuscript:

P6§1: "The demonstrated concept offers a wide pulse shaping window w.r.t. pulse duration, photon energy and more complex phase shapes. In particular, the recent progress in echo-enabled harmonic

⁷ A. A. Lutman, G. Penco, P. Craievich, and J. Wu, Impact of an Initial Energy Chirp and an Initial Energy Curvature on a Seeded Free Electron Laser: Free Electron Laser Properties, *J. Phys. A: Math. Theor.* 42, 085405 (2009).

⁸ M. Beye, M. Gühr, I. Hartl, E. Plönjes, L. Schaper, S. Schreiber, K. Tiedtke, and R. Treusch, FLASH and the FLASH2020+ Project—Current Status and Upgrades for the Free-Electron Laser in Hamburg at DESY, *Eur. Phys. J. Plus* 138, 193 (2023). S. Reiche, Seeding at SwissFEL, in *X-Ray Free-Electron Lasers: Advances in Source Development and Instrumentation VI*, Vol. 12581 (SPIE, 2023), pp. 16–23.

B. Liu et al., The SXFEL Upgrade: From Test Facility to User Facility, *Applied Sciences* 12, 1 (2022).

⁹ P. Finetti et al., Pulse Duration of Seeded Free-Electron Lasers, *Phys. Rev. X* 7, 021043 (2017).

¹⁰ D. Gauthier et al. *Phys. Rev. Lett.* 115, 114801 (2015)

¹¹ P. K. Maraju et al., *Nature* 578, 7795 (2020).

generation [61,62] promises to extend the pulse shaping concept to the soft X-ray domain (≈ 600 eV) where localized core-electron states can be addressed.”

Moreover, the following statements in our manuscript support our points:

P6§1: “The generation of coherent attosecond pulse trains, with independent control of amplitude and phases has been demonstrated at seeded FELs [43], bringing pulse shaping applications on the attosecond time scale within reach.”

2.2

The authors mention potential implications of pulse shaping techniques for X-rays several times. It would be valuable to assess the feasibility of such techniques with respect to current operational XFELs. If not feasible, this should be explicitly stated, considering factors such as the required pulse intensity and potential competing ionization processes.

We agree with the reviewer that our statements require clarification w.r.t the soft and hard X-ray domain. As outlined in comment 2.1, pulse shaping-based coherent control in the soft X-ray regime is realistic in the near future at several FEL facilities. While the dynamic developments in accelerator physics and seeded FELs may enable pulse shaping applications even in the hard X-ray domain in the further future (e.g. with cascaded operation of EEHG, as applied in HGHG¹²), we refrain from claims in this direction at the current state of technology.

Regarding the feasibility of applications in the soft X-ray domain: Competing ionization is a major obstacle in both spectroscopy and imaging applications in this spectral domain. A lot of effort is dedicated to solving this problem. However, solutions based on coherent control have hardly been explored, so far. First theoretical work shows how quantum optimal control in the XUV/X-ray domain can significantly improve target state preparation while suppressing background ionization^{13,14} or improve the detection to selectively enhance weak signals of interest^{15,16}. Our work provides the first experimental demonstration in this direction, which is paramount to foster more experimental and theoretical efforts exploring the new possibilities unfolded by this research direction.

To account for the reviewer’s comment, we have changed the statements in our manuscript to avoid confusion between the soft and hard X-ray domain and added the following statement:

P6§1: “Besides, selective control schemes may reduce the influence of competing ionization processes ubiquitous in XUV/X-ray spectroscopy and imaging experiments, where our work provides the first experimental demonstration in this direction.”

2.3

It seems that a slightly different resonance transition was selected for this experiment compared to a similar previous experiment [26]. Was there a specific rationale behind choosing this transition, apart from the increased transition dipole moment? Considering that the other resonance ($1s2 \rightarrow 1s4p$) is

¹² E. Allaria et al., Nature Photon 7, 913 (2013).

¹³ L. Greenman et al., Phys. Rev. A 92, 013407 (2015).

¹⁴ U. Saalman et al., Phys. Rev. Lett. 121, 153203 (2018).

¹⁵ R. E. Goetz et al., Phys. Rev. A 93, 013413 (2016).

¹⁶ D. Keefer and S. Mukamel, Phys. Rev. Lett. 126, 163202 (2021).

very close (within 2 eV) and represents an identical system, was there an effort made to replicate the results previously reported [26]?

The aim of our study was the demonstration of strong-field quantum control in the XUV domain and the probing of dressed continuum states. To introduce these new concepts, we chose a clean test system, that is a two-level system well isolated from other optically active states. In particular, studies in the NIR wavelength regime have shown that strong-field coherent control becomes considerably more complex, if more than one optical transition is involved¹⁷. For the $1s^2 \rightarrow 1s2p$ resonance, the closest optically active state is the $1s3p$ at +1.87eV. In contrast, the $1s4p$ state is embedded in the He Rydberg progression ($2s3p$ at -0.65eV, $1s5p$ at +0.31eV, $1s6p$ at +0.47eV,...) and, thus, the energetically much closer states may influence the dynamics of the system. As another aspect pointed out by the reviewer, the dipole moment of the $1s^2 \rightarrow 1s2p$ resonance is larger than the one of the $1s^2 \rightarrow 1s4p$ transition (by a factor of 2.7). Due to the larger dipole moment, the Rabi dynamics become directly observable in the raw photoelectron data which reduces uncertainties in the data analysis. For these reasons we focused our study on the $1s^2 \rightarrow 1s2p$ resonance. The $1s^2 \rightarrow 1s4p$ resonance was not investigated in the limited time available during our beamtime. Thus, we cannot conclude about the dynamics expected for this transition. However, Nandi et al. seem to observe an influence of nearby states on the photoionization dynamics. Hence, it would be indeed interesting to apply our quantum control method to the $1s^2 \rightarrow 1s4p$ transition in the future.

To make our choice clearer to the reader, we added the following statement to the manuscript:

P5§2: “we study the energetically well-isolated transition $1s^2 \rightarrow 1s2p$, where contributions from neighboring optically active states should be negligible. This provides us a clean two-level system and greatly simplifies the data interpretation.”

2.4

Is the plot Fig2a for a flat phase φ ? Is the asymmetry present here between the photoelectron yield from the lower energy dressed state and the higher energy dressed state the same as the one referred to later at GDD = 0 fs² (Fig. 4c)? I realize that the uncertainty in the GDD may be large.

The data in Fig. 2a was taken for GDD = +135 fs². At this level of chirp, the asymmetry in the photoelectron distribution is similar to the case of GDD = 0 fs² (see Fig. 4c). We note, that this data set was taken for a photon energy of 21.26 eV, thus blue-detuned by 40 meV w.r.t. the $1s^2 \rightarrow 1s2p$ resonance, which also affects the symmetry of the photoelectron distribution. The uncertainty in the GDD is estimated to be within +/- 100 fs² (see comment 1.2).

We added the missing information to the manuscript.

2.5

Figure 2a suggests a decrease in the energy bandwidth of the photoelectron distribution from left to right. Could the authors explain the reason for this?

Our data analysis indicates that this effect is due to the focal intensity average in the experiment. Figure 3 in the revised manuscript now shows the calculations including the focal intensity average and for a single intensity. While for a single intensity, the width of both AT peaks seem identical, for

¹⁷ M. Krug et al. New J. Phys. 11, 105051 (2009).

the intensity-averaged data, the width of the lower energy peak is broader than the width of the higher-energy peak.

2.6

In Fig. 3, How should one understand the emergence of a small peak for GDD values = 135 fs², -591 fs², -1127 fs² at around eKE = 18.1 eV? If I understand correctly, any excitation due to focal averaging was subtracted out from these figures? Also, why does this peak not appear in the simulation?

This peak is due to the residue from the subtraction of the lower intensity contributions within the focal intensity average. For the subtraction, we extrapolate the data set in Fig. 2c to match the experimental parameters under which the data in Fig. 3 and 4 were recorded. The uncertainty in this procedure leads to a small residue. However, this uncertainty does not compromise the interpretation of our results.

To make this clearer we added the following statement in the caption of Fig. 3: "The small peak at 18.13 eV is due to imperfect removal of the lower intensity contribution from the aberrated focus."

2.7

Regarding the asymmetry in the AT doublet spectrum, simulations with photoionization interference channels turned off still result in asymmetry. It appears that this is the basis for saying the mechanism for the asymmetry in the AT doublet spectrum at play in Nandi et al.[26] is not manifesting here. The authors mention that "For helium such effects are, however, expected in a narrow parameter range [26, 58], which lies outside the regime probed in our experiment." Could the authors offer an intuitive explanation for why these interference effects are absent in this experiment?

The 1s2p state is much better isolated from neighboring states than the 1s4p state (see comment 2.3). Therefore, photoionization interference channels due to near-resonant states are negligible for the 1s2p state. To confirm this, we performed a test calculation where we suppressed ionization paths via neighboring states and only the resonant ionization path via the 1s2p state is present. We still observe an asymmetry in the photoelectron distribution for this case. Moreover, the overall ionization yield is basically identical. From this we conclude, that ionization pathways via neighbouring states are negligible and do not explain the asymmetry observed in our experiment. We note that, Nandi et al. studied a different state and worked under different experimental conditions. In particular, Nandi et al. used linear laser polarization where the asymmetry effect caused by the coupling to the ionization continuum is much weaker. Therefore, our results do not necessarily contradict their work.

We have added the mentioned calculation to the Supp Info of the manuscript (see Supp. Info. IV).

We agree that the statement "For helium such effects are, however, expected in a narrow parameter range [26, 58], which lies outside the regime probed in our experiment." is misleading and we therefore removed it. In addition, we added the following text:

P5§2: "For confirmation, we performed a calculation with a modified model in which any two-photon ionization via near-resonant states (except for the 1s2p state) was suppressed, and, thus, possible

photoionization interference effects are eliminated. Still, we observe a pronounced asymmetry in the AT doublet amplitudes (see Supp. Info. IV).”

Minor suggestions:

1. It would be useful if the authors briefly describe and provide an idea of group delay dispersion since it is the main parameter for quantum control. Perhaps specify the incident classical vector potential of the pulse with the GDD parameter in Sec.III. B.

We have added this information following to the reviewer’s suggestion in Sec. III B.

Response to referee #3 :

The manuscript by Richter et al. reports on a strong-field quantum control experiment achieved by pulse shaping in the EUV. The phase of the EUV pulses is achieved through intense laser seeding of the FEL. When the phase shaped pulse interacts with He atoms, the strong-field causes Autler-Townes (AT) energy splitting of an intermediate state (shown in Fig. 2). The population of the AT states is followed by the kinetic energy of the electrons following ionization. When the phase curvature is negative (positive) the lower (upper) state is preferentially populated (shown in Figs. 3 and 4).

I consider these results highly significant because they open the door to experiments with tailored spectral phases in the EUV. The presentation is clear, and the figures are high quality. The results support the claims in the abstract and the conclusions.

While the paper can be published as is, I suggest the authors consider the following:

3.1

Control of the curvature of the spectral phase opens the door for experiments along the lines of Silberberg and Dantus, where a sinusoidal function or a combination of second and third order spectral phase can be used to selectively excite two- and three-photon transitions. Similarly, spectral phase control has been used by Silberberg and Dantus to drive selectively stimulated-Raman transitions. These approaches are described in the following reviews [10.1146/annurev.physchem.040808.090427; 10.1002/cphc.200400342; Ref 32 in this manuscript, and references within these review articles] The advantage of these approaches to coherent control is that they are highly reproducible and have led to applications. In contrast, few if any experiments using genetic algorithms have been reproduced.

We thank the reviewer for pointing out these references and for pointing out prospective applications of our work. We have added the suggested references (see p2§1) and removed the less appropriate reference (Ref. 14 in initial submission): R. J. Levis, G. M. Menkir, and H. Rabitz, “Selective bond dissociation and rearrangement with optimally tailored, strong-field laser pulses”, Science 292, 709–713 (2001)

Indeed, an extension to sinusoidal or cubic phase modulation is well-feasible with our approach, which would enable many more quantum control experiments, as suggested by the reviewer. Accordingly, we added the following statement to the conclusions:

P6§1: “The demonstrated scheme already sets the basis for highly efficient adiabatic population transfer [1, 2] and an extension to cubic or sinusoidal phase shaping would open-up many more interesting control schemes [32, 34].”

Comment by the editor:

In addition, please also elaborate on extending your method to shorter wavelengths using simulations or models (as experiments may not be feasible at this stage).

Please see our answer to comment 2.1.

Other changes made in the manuscript:

- Update of Ref. 59 which is now published in Phys. Rev. A.
- Added the following funding in the acknowledgements:
DFG project STI 125/24-1
COST Action CA21101